# Host Cell Redox Alterations Promote Latent HIV-1 Reactivation through Atypical Transcription Factor Cooperativity

**DOI:** 10.3390/v14102288

**Published:** 2022-10-18

**Authors:** Emily Cruz-Lorenzo, Nora-Guadalupe P. Ramirez, Jeon Lee, Sonali Pandhe, Lei Wang, Juan Hernandez-Doria, Adam M. Spivak, Vicente Planelles, Tianna Petersen, Mamta K. Jain, Elisabeth D. Martinez, Iván D’Orso

**Affiliations:** 1Department of Microbiology, The University of Texas Southwestern Medical Center, Dallas, TX 75390, USA; 2Lydia Hill Department of Bioinformatics, The University of Texas Southwestern Medical Center, Dallas, TX 75390, USA; 3Department of Pharmacology, The University of Texas Southwestern Medical Center, Dallas, TX 75390, USA; 4Cecil H. and Ida Green Center for Reproductive Biology Sciences, The University of Texas Southwestern Medical Center, Dallas, TX 75390, USA; 5Division of Infectious Diseases, Department of Medicine, University of Utah, Salt Lake City, UT 84112, USA; 6Department of Pathology, University of Utah, Salt Lake City, UT 84112, USA; 7Department of Internal Medicine, The University of Texas Southwestern Medical Center, Dallas, TX 75390, USA; 8Parkland Health & Hospital System, 5200 Harry Hines Blvd, Dallas, TX 75235, USA

**Keywords:** HIV-1, transcription factors, cell signaling, latency, reactivation, latency reversing agents

## Abstract

Immune cell state alterations rewire HIV-1 gene expression, thereby influencing viral latency and reactivation, but the mechanisms are still unfolding. Here, using a screen approach on CD4^+^ T cell models of HIV-1 latency, we revealed Small Molecule Reactivators (SMOREs) with unique chemistries altering the CD4^+^ T cell state and consequently promoting latent HIV-1 transcription and reactivation through an unprecedented mechanism of action. SMOREs triggered rapid oxidative stress and activated a redox-responsive program composed of cell-signaling kinases (MEK-ERK axis) and atypical transcription factor (AP-1 and HIF-1α) cooperativity. SMOREs induced an unusual AP-1 phosphorylation signature to promote AP-1/HIF-1α binding to the latent HIV-1 proviral genome for its activation. Consistently, latent HIV-1 reactivation was compromised with pharmacologic inhibition of oxidative stress sensing or of cell-signaling kinases, and transcription factor’s loss of expression, thus functionally linking the host redox-responsive program to viral transcriptional rewiring. Notably, SMOREs induced the redox program in primary CD4^+^ T cells and reactivated latent HIV-1 in aviremic patient samples alone and in combination with known latency-reversing agents, thus providing physiological relevance. Our findings suggest that manipulation of redox-sensitive pathways could be exploited to alter the course of HIV-1 latency, thus rendering host cells responsive to help achieve a sterilizing cure.

## 1. Introduction

Human immunodeficiency virus (HIV-1) establishes long-lived latent infections in CD4^+^ T cells [1,2] and other cell types that compromise its eradication. These latent reservoirs harbor silent and lowly expressed yet inducible, replication-competent and -incompetent proviruses that are refractory to current antiretroviral therapy (ART) and to immune surveillance. Cessation of ART thus results in rapid viral rebound, leaving patients to remain on lifelong therapy. As such, obtaining a comprehensive understanding of HIV-1 latency establishment, maintenance and reactivation will facilitate the development of new therapeutics and possibly a sterilizing cure.

HIV-1 integration [3] allows the virus to persist indefinitely in host cells and be transcriptionally regulated by a multitude of host and viral mechanisms. Transcriptionally, HIV-1 is regulated at two main levels, referred to as the “host” and “viral” phases [4]. During the “host” phase, ligands (e.g., antigen, cytokines) from the immune microenvironment, environmental stress, and other cell-extrinsic signals (e.g., hormones and growth factors) trigger signaling events through mitogen-activated protein kinases (MAPKs) [5]. Activated kinases then phosphorylate master transcription factors (TFs) to activate them through either translocation from the cytoplasm into the nucleus (e.g., NF-κB and NFAT) and/or through protein synthesis and site-specific phosphorylation (e.g., activator protein 1 [AP-1]). Activated TFs then bind to their cognate cis-elements at the 5′ long terminal repeat (5′-LTR) of the HIV-1 proviral genome [6,7,8,9,10,11]. This binding event allows for the assembly of the transcription machinery to initiate the synthesis of a low number of viral transcripts, enabling the production of trans-activator of transcription (Tat), which facilitates a positive feedback loop through the recruitment of host machinery to the 5′-end of viral nascent pre-mRNAs (reviewed in [12]). As such, it is the sequential activation of host and viral phases that facilitates efficient latent HIV-1 reactivation.

Given the broad utilization of these pathways by immune cells, it became evident that latent HIV-1 can be reactivated by various external stimuli [13,14]. One therapeutic strategy to purge the latent reservoir is known as “shock and kill”, where latent HIV-1 is first shifted into a state of productive synthesis using latency-reversing agents (LRAs), and later eliminated by the host immune system or by cytopathic effects. Several strategies are currently employed to initiate latent HIV-1 proviral reactivation, including immunotherapy, T-cell receptor (TCR)-mediated signaling in a polyclonal fashion and LRAs targeting pathways other than T-cell activation [14,15]. For example, proinflammatory cytokines (e.g., TNF-α, CD154) induce latent HIV-1 transcription in CD4^+^ T cells from ART-treated patients [14]. Protein kinase C (PKC) agonists such as phorbol esters (PMA), potently induce T-cell activation and HIV-1 transcription via activation of ERK signaling and NF-κB [13,16], but PMA has oncogenic properties making it unsuitable for clinical trials [14,17,18].

Additionally, a large diversity of LRAs reactivating latent HIV-1 from cell models of latency has been described, including histone deacetylase (HDAC) inhibitors (e.g., SAHA), and epigenetic competitors (mimics) of bromodomain (BRD)-containing proteins (e.g., JQ1 and iBET) [19,20]. While these strategies have shown promise in vitro, many ex vivo and in vivo studies have determined that these treatments, especially as single agents, inefficiently activate latent reservoirs, resulting in their minimal reduction [21]. Together, this implies that the current pipeline for small new compounds and chemistries against latent HIV-1 remains narrow. As such, the need to develop new therapeutics or LRAs to selectively reactivate latent HIV-1 has become urgent. In a unique cell-based high-throughput screen targeting latent herpesvirus (CMV), the small compound JIB-04 was found to epigenetically reactivate a silent viral locus [22,23] by potentially inhibiting the activity of a large family of histone demethylases (HDMs) known as Jumonji [23,24,25].

Here, we used JIB-04 analogs from the NCI diversity library [26,27], referred to as SMOREs (Small Molecule Reactivators), to identify leads capable of reactivating latent HIV-1. SMOREs fulfill the criteria of small molecules because they are low-molecular-weight (≤1000 Da, size of 1 nm) organic compounds that have the potential of being metabolized into smaller products promoting cell-sensing and -signaling events and/or engaging with host cell targets to regulate biological processes. SMOREs have numerous advantages as potential therapeutics since they have not been previously examined nor mechanistically characterized for the study of latent HIV-1 reactivation for future translational approaches. SMOREs are chemically distinct from previously known epigenetic modulators such as HDAC inhibitors, PKC agonists, and epigenetic mimics [14,28]. SMOREs comprise seven different chemical structure families (EPH, UIBK, MK, INHE, BABA, EJMI) and a handful of singlets and are structurally related by the presence of hydrazone and benzene groups (Figure 1a).

We used the SMORE library in a targeted screen to probe their function in the context of latent HIV-1 to identify leads for both mechanistic studies and preclinical applications. We illustrate the scientific and potential clinical value of these novel chemical modulators. SMOREs showed an intermediate level of potency as compared with T-cell activators and functioned with other epigenetic mimics to provide combinatorial activation in patient-derived resting CD4^+^ T cells in the presence of ART. Mechanistically, SMOREs altered the CD4^+^ T-cell state and consequently promoted latent HIV-1 gene expression through a previously unknown redox-sensing and transduction program composed of signaling kinases (MEK-ERK axis) and TF (AP-1 and hypoxia-inducible factor 1-alpha [HIF-1α]) cooperativity.

## 2. Results

### 2.1. A Small Compound Screen Identifies a Novel Class of HIV-1 Latency-Reversing Agents

To discover SMOREs with latency-reversal potential, we first performed a targeted screen and then assessed candidate’s potency, efficacy, and selectivity towards the latent HIV-1 genome. To identify leads that are not unique to a particular latency model, SMOREs were initially tested on the J-Lat 10.6 cell model expressing GFP [29] at 1 μM (and a final DMSO concentration of 0.1%) for 24 h (Figure 1a), and later validated in the E4 cell model [30] (Appendix A). Latent HIV-1 reactivation was assayed by measuring percentage of GFP-positive cells by flow cytometry. SMOREs yielding a 3-fold or greater increased GFP expression relative to negative control were considered for further studies. To rule out false-positive results, SMOREs were tested in parallel in parental Jurkat T cells, which showed no signal over background (DMSO-treated cells) (data not shown).

Based on these criteria, a narrow set of candidates (EPH294, EPH334, EJMI02, JIB-04, SMORE14, and SMORE53) reproducibly reactivated latent HIV-1 in J-Lat 10.6 (Figure 1a), and E4 cells (Appendix A) by ~3-to-10-fold increased GFP expression over background. In addition to activating J-Lat 10.6 and E4 (Appendix A), 5 compounds (EPH294, EPH334, EJMI02, JIB-04, and SMORE53) also induced other J-Lat cell models of latency (15.4, 9.2 and 6.3) harboring a single copy of the HIV-1 genome tagged with GFP and integrated in unique locations of the host (Appendix A–e); however, some models (8.4) remained refractory to SMORE treatment (Appendix A), potentially attributed to their placement in highly heterochromatic regions of the genome. These results demonstrate that SMOREs reactivate latent HIV-1 placed in 5 different euchromatic regions, albeit at various levels, and thus have broad latency-reversal potential.

Since the screen was performed at relatively high concentrations (1 μM), we then performed standard dose–response studies in 10.6 cells to calculate potency (EC_50_ values of latent viral reactivation by flow cytometry), with the top five candidate SMOREs described above, and used TNF-α and JQ1 as positive controls. As a proof of concept, we compared the dose–response between the active form of JIB-04 (E isomer) [23], the inactive form (Z isomer), and vehicle (DMSO), which revealed ~2–10% reactivation with JIB-04 (E) and no substantial stimulation by JIB-04 (Z) for the concentrations tested (0–10 μM) (Figure 1b). Candidate SMOREs showed a broad range of potency (Figure 1c). Additionally, we calculated efficacy (level of maximum activation achieved) using flow cytometry and RT-qPCR. The five SMOREs activated the same percentage of cells or even a 2–3–fold higher percentage of cells than JQ1 and yielded similar fold changes in HIV-1 expression.

Because some LRAs cause latent HIV-1 reactivation upon induction of apoptosis [31], we next tested whether SMOREs triggered cell death. J-Lat 10.6 cells were treated with the top five candidates, as in the screen-based approach, and incubated with Ghost Dye Violet (450), which is a free amine-reactive viability dye used to discriminate viable from nonviable cells using flow cytometry. None of the compounds showed any significant cell death (percentage cell viability ranged between 94–99%, depending on the small compound, relative to vehicle (DMSO) reference control) (Figure 1c); results were cross-validated with Trypan Blue staining.

Collectively, we discovered a new set of small compounds that reactivate latent HIV-1 in cell-based models independent of integration-site placement. Because the screen yielded five small compounds with similar activation profiles, for subsequent mechanistic studies we focused on EPH334 as the most potent and efficacious SMORE.

### 2.2. SMOREs Induce HIV-1 RNA Processing and Function Combinatorially with Epigenetic LRAs

The previous data showed that SMOREs mediated reactivation of latent HIV-1, as judged by the increase in the percentage of cells expressing GFP by flow cytometry (Figure 1). However, this analysis does not inform us of: (i) whether transcription arises from either the normal activity of the HIV-1 promoter (which generates fully elongated, mature transcripts) or from host gene promoter-initiated activity (which is known as read-through transcription and generates aberrant, immature transcripts); and (ii) whether HIV-1 RNA is properly processed after transcription. Precise maturation of HIV-1 RNA (namely transcription elongation, splicing and 3′-end processing) [32] is needed to generate the correct ratio of viral structural and regulatory products to perpetuate the infection. During HIV-1 transcriptional activation, a collection of at least three types of transcripts is generated: (1) an unspliced (US) ~9 Kb-long viral RNA form encoding the structural proteins Gag and Pol, (2) a singly spliced (SS) ~4.5 Kb-long RNA lacking the coding sequences for Gag and Pol, and (3) several multiply spliced (MS) ~1.8 Kb-long transcripts (Figure 2a).

To test whether SMOREs induce host or HIV-1 promoter-initiated transcription and viral RNA processing, we treated 10.6 cells with or without EPH334 and performed RT-qPCR assays with primer pairs that selectively amplified the different RNA processed forms (US, SS, and MS), and observed that EPH334 induced the expected splicing pattern of the viral pre-mRNA (Figure 2b), at least with the major splicing acceptor and donor sites tested. EPH334 also induced 3′-end processing and polyadenylation, as revealed by the increase in synthesis of the fully mature RNA form containing a poly(A)-tail [33] (Figure 2b). Together, our screen and HIV-1 RNA processing assay allowed us to identify lead compounds that stimulate LTR promoter activity in contrast to compounds showing abortive and aberrant host–HIV-1 chimeric transcripts not indicative of productive HIV-1 gene expression and processing.

Because it is unlikely that a single compound treatment may be effective in the clinics, we evaluated synergy between SMOREs and known epigenetic modulators to identify combinations that can yield a significantly greater response. As proof of concept, we first selected JQ1, which has been proposed to partly activate HIV-1 transcription by binding to and competing off the BRD-containing BRD4 protein from acetylated chromatin at the proviral LTR [19,34,35,36]. For this assay, we used two EPH334 doses (0.5 and 1 μM) alone or with JQ1 max activation dose (0.5 μM). In combination with JQ1, EPH334 achieved higher reactivation levels (Figure 2c), at concentrations that do not significantly induce cytotoxicity (Figure 2d), relative to the individual components alone. In an “order of addition” experiment (Figure 2e), pretreatment with JQ1 enhanced EPH334-mediated latent HIV-1 reactivation in 10.6 cells and decreased cytotoxicity (Figure 2f,g).

To precisely study the kinetics of latent HIV-1 reactivation in comparison with other LRAs and drug combinations, we treated 10.6 cells with the EPH334, JQ1, and EPH334+JQ1 combination, and collected cells at different time points post-treatment (1, 2, 4, 8, and 20 h) to monitor kinetics of activation by RT-qPCR. EPH334 induced HIV-1 RNA synthesis with kinetics consistent with transcription activation mechanisms and EPH334+JQ1 combination was more effective than the sum of the individual treatments (Figure 2h). Combination of EPH334 and JQ1 rapidly activated HIV-1 gene expression and to higher levels compared to EPH334 or JQ1 alone, again revealing their additive effect.

Given that the above experiments have been performed in immortalized cell models of latency, and because these models may not fully recapitulate mechanisms governing latency in vivo, we tested whether EPH334 disrupts latency ex vivo from aviremic patient samples alone, and in combination with JQ1 to provide evidence of utility in physiologic models of disease. For this, we enrolled patients with suppressed viremia and normal CD4^+^ T-cell counts (<50 HIV-1 mRNA copies/mL of plasma, patients were under ART regimen for at least 6 months) for blood draws (see details in Methods). Peripheral blood mononuclear cells (PBMCs) were isolated by a standard density gradient procedure and resting CD4^+^ T cells (rCD4^+^) were purified following established protocols [37]. CD4^+^ T cell purity relative to PBMCs was determined by flow cytometry of the purified sorted cells for the expression of CD4^+^ and the lack of activation markers using specific conjugated antibodies (anti-CD4, -CD69 and -CD25) in addition to isotype controls (Appendix A).

Once rCD4^+^ T-cell purity was confirmed, cells (5 × 10^6^ cells per condition) from four donors were treated ex vivo in the presence of ART (Raltegravir) with EPH334, JQ1, EPH334 plus JQ1, and anti-CD3/CD28 antibodies as positive control for 48 h to measure latent HIV-1 reactivation (using Taqman to quantitate released viral genomic RNA copies to the supernatant using the REVEAL (rapid ex vivo evaluation of antilatency) assay [37]) (Figure 2i). Several conclusions can be raised from this experiment. First, JQ1 modestly activated HIV-1 in all four patients tested. Second, three out of four patients responded well to the increased EPH334 dose, albeit with donor-to-donor variability. Third, the patient that did not respond well to EPH334 alone showed robust stimulation in combination with JQ1. Fourth, one patient responded potently to EPH334 alone but not in combination with JQ1. The exact reasons explaining this unexpected behavior remain unclear. Fifth, the combination of EPH334 and JQ1 reactivated HIV-1 additively (*p* < 0.05) at levels comparable with anti-CD3/CD28 treatment. Importantly, cell viability remained largely unaffected by any of the treatments (Figure 2j). Collectively, this analysis indicated SMOREs could have clinical utility, especially in combination with other known LRAs.

### 2.3. Transcriptome Profiling Unveils Induction of Host Cell Master Transcriptional Regulators in Response to Oxidative Stress Cell Signaling

To start defining how our top SMORE (EPH334) functions to reactivate latent HIV-1, we performed transcriptome profiling using ribo-minus RNA-seq [38] to first predict and then validate cell processes linked to HIV-1 proviral fate alterations. To obtain a temporal view of host cell gene expression changes, we collected three time points of EPH334 treatment: early (1 h), intermediate (8 h), and late (16 h); performed bulk RNA-seq for each time point in triplicate to increase statistical robustness; and identified differentially expressed genes (DEGs) with a false-discovery rate [FDR] q-value < 0.05 by comparing transcriptomes in each time point of EPH334-treated cells to an untreated control (Figure 3a and Appendix A). Gene ontology (GO) analysis was performed to identify hallmark gene sets functionally enriched among the most up- and downregulated genes using Enrichr [39] and Metascape [40] to reveal programs involved in latent HIV-1 reactivation (Appendix A).

In the early time point, 43 DEGs (26 upregulated and 17 downregulated) were found with log_2_fold change (FC) > 1, and 118 DEGs (60 upregulated and 58 downregulated) with log_2_FC > 0.5 (Figure 3a and Appendix A). In the later time points, the DEGs largely increased, suggesting the activation of transcriptional cascades, in which TFs are induced earlier to stimulate cell programs at later time points [41]. Importantly, we validated a subset of the DEGs identified through RNA-seq using RT-qPCR assays (Appendix A).

Throughout the previously described time course, we noted certain phenotypes of interest. Surprisingly, genes associated with “detoxification of inorganic compound (GO:0061687)” (*p* = log_10_−11.84), “stress response to metal ions (GO:0097501)” (*p* = log_10_−11.84) and “cellular responses to external stimuli (GO: R-HAS-8953897)” (*p* = log_10_−5.163), were enriched among the most upregulated (Figure 3b and Appendix A). This gene set was primarily composed of a family of metallothioneins (MTs) whose expression peaked at 8 h post-EPH334 treatment (e.g., MT1E log_2_FC = 5.99; MT1X log_2_FC = 4.59). MTs are antioxidant proteins involved in metal binding and detoxification for protection against oxidative stress by shifting the redox balance towards reduction [42,43], potentially indicating that EPH334 triggers oxidative stress by being metabolized into reactive radicals, which then generate reactive oxygen species (ROS) (see Figure 4 below).

Metascape-enriched gene clusters showed significant upregulation of metal detoxification genes at the early time point, consistent with activation of a detoxification response pathway (Figure 3c). In agreement, oxidative stress-responsive TFs of the EGR and AP-1 families, which are also known to be induced upon cell signaling to stress conditions [44], were also upregulated (e.g., *EGR1* log_2_FC = 3.47 and *FOS* log_2_FC = 1.71) (Appendix A). We also observed several target genes of HIF-1α (16 out of 83, Adjusted *p* = 2.41 × 10^6^, TRRUST Transcription Factors 2019 Database) upregulated (e.g., *CA9* and *PFKFB4*), indicative of enhanced HIF-1α transcriptional activity at 8 and 16 h (Figure 3b,c and Appendix A). During the late time point, most upregulated genes were associated with increased glycolytic metabolism, consistent with a HIF-1α–mediated transcriptional response [45].

Gene sets associated with processes regulating chromatin condensation, including “Histone Deacetylases (HDACs) deacetylate histones (GO: R-HSA-3214815)”, “RNA Polymerase I Promoter Opening (GO: R-HSA-73728)”, and “Arginine Methyltransferases (RMTs) methylate histone arginine’s (GO: R-HSA-3214858)”, were enriched in downregulated genes throughout all time points evaluated (Figure 3b,c and Appendix A) including histone genes (Appendix A), consistent with the idea that detoxification and DNA damage responses induce downregulation of histone gene expression [46] (Figure 3b,c).

An unwanted effect of latency reversal is T-cell activation. While EPH334 induces markers of T-cell activation (e.g., *CD69*, *CD25*) in J-Lat 10.6 cells (Appendix A) and resting CD4^+^ T-cells treated ex vivo (Appendix A), their induction was ~10-fold lower compared with agents eliciting potent T-cell activation, such as PMA/Ionomycin.

Collectively, the transcriptome analysis predicts that EPH334 triggers early processes such as ROS, which is known to induce key master transcriptional regulators (AP-1 and HIF-1α) that have the potential to bind the HIV-1 LTR to activate latent proviruses [47,48,49]. Alterations in redox may also activate host cell processes (including hypoxia and T cell differentiation and activation) and inactivate the activity of HDACs, potentially explaining the downregulation of a subset of Pol II and Pol I genes (Figure 3b,c). Below we focus on the mechanism of EPH334-mediated cell signaling and transcriptional regulation leading to latent HIV-1 reactivation.

### 2.4. EPH334 Induces Oxidative Stress to Stimulate Latent HIV-1 Reactivation in CD4^+^ T Cell Models of Latency

Small molecules such as EPH334 can bind target molecules in host cells and/or be metabolized, eliciting specific biological responses. Using the transcriptome data suggesting activation of host cell stress responses, we inferred a model whereby EPH334 is metabolized into reactive radicals to elicit ROS (Figure 4a), which occurs when the steady-state balance between pro- and antioxidants in the cell changes through the generation of free radicals [50]. ROS can then produce a variety of cellular effects, including: (1) oxidative stress, consequently activating stress-related MAPKs involved in the synthesis of redox-sensitive master TFs (e.g., AP-1); and (2) hypoxia mimicry, consequently activating a hypoxia-responsive program by hypoxia-inducible factors such as HIF-1α (Figure 4a). If this model is correct, we would then expect a temporal series of events in which ROS accumulation is triggered, first leading to the activation of redox-sensitive kinases (e.g., ERK1/2 and p38), and then of redox-sensitive TFs that can directly reactivate latent proviruses in addition to host target genes (Figure 4a).

To directly test the various parts of this model, we first subjected J-Lat 10.6 cells to a temporal EPH334 treatment and measured ROS production using the fluorogenic reporter CellROX Green dye. EPH334 elicited a very early (noticeable as early as 7.5 min), time-dependent, and statistically significant increase in ROS production (Figure 4b), which was comparable (~2-fold lower) to excess amounts of a well-known ROS inducer (Menadione) (Appendix A). As negative control, EPH334 treatment (2 h) of unstained J-Lat 10.6 cells produced no GFP signal derived from latent HIV-1 reactivation (Appendix A), indicating that the signal measured is from ROS generation detected by the CellROX Green dye. Additionally, cells treated with EPH334 in the absence of CellROX Green dye did not yield any autofluorescence (Appendix A), strongly indicating the specificity of the measured signal.

To test if EPH334-induced ROS production promoted latent HIV-1 reactivation, J-Lat 10.6 cells were preincubated for 2 h with the ROS scavenger N-Acetylcysteine (NAC, 10 mM) or vehicle control (tissue culture media) prior to stimulation with EPH334 (Figure 4c). To first validate ROS inhibition, NAC-pretreated cells were stimulated with EPH334 for 1 h (or vehicle control DMSO) and stained with CellROX Green to assess the inhibition of ROS formation by flow cytometry. Importantly, NAC, but not vehicle control, reduced EPH334-induced ROS production by ~90% (Figure 4d) and functioned in a dose-dependent manner (Appendix A).

Having established the experimental system, we then measured latent HIV-1 reactivation after 8 h of EPH334 stimulation by RT-qPCR and found that NAC reduced (~2-fold) EPH334-induced ROS-mediated HIV-1 gene expression in a statistically significant manner (*p* < 0.05, unpaired Student’s *t*-test) (Figure 4e). NAC and EPH334 co-treatment did not reduce cell viability, as the percentage of live cells did not significantly change throughout the experimental time course, as revealed by Trypan Blue staining (Figure 4f).

To extend these data to more physiologically relevant systems, we used the primary model of latency in central memory T cells (T_CM_) [51], one of the most frequent cell types harboring latent HIV-1. We isolated PBMCs from healthy donors to purify naïve CD4^+^ T cells using negative selection, which were then polarized with cytokines (IL-2, TGF-β) and neutralizing antibodies (anti–IL-4 and anti–IL-12), and then infected with replication-defective HIV-1 (pNL4.3-delta*Env*-nLuc-2A*Nef*-VSVG). Once T_CM_ transitioned to a resting memory phenotype, as judged based on the presence of CD45RO^high^, CD25^low^, and CD69^low^, [52] cells were then treated with EPH334 or vehicle DMSO control, in the absence and presence of NAC. Notably, EPH334 induced latent HIV-1 reactivation, which was partially compromised by the addition of NAC (Figure 4g). As with the 10.6 model of latency, NAC and EPH334 cotreatment did not significantly alter cell viability (Figure 4h).

Together, the data suggest that EPH334-induced ROS production partly mediates latent HIV-1 gene expression, which is potentially consistent with recent reports linking antioxidative stress responses with latent HIV-1 reactivation [53].

### 2.5. EPH334 Stimulates Redox-Sensitive TFs That Mediate Latent HIV-1 Reactivation

A hallmark of oxidative stress signaling is the synthesis of immediate early genes (IEGs), encoding for master transcriptional regulators activating transcriptional cascades (e.g., HIF-1α as well as AP-1 and EGR family members). Given that EPH334 induced ROS production to reactivate latent HIV-1 (Figure 4), we reasoned that oxidative stress-responsive TFs required for latent HIV-1 reactivation may be induced (Figure 4a). Supporting this notion, transcriptional profiling by RNA-seq revealed that EPH334 triggered a rapid and time-dependent increase in gene expression of members of the AP-1 family of TFs such as *FOS* and *JUN* (Appendix A), which are known to form the canonical AP-1 (c-Fos/c-Jun) heterodimer to transactivate the HIV-1 provirus [8]. RNA-seq validation by RT-qPCR revealed that *FOS* and *JUN* were indeed temporally induced in J-Lat 10.6 cells in response to EPH334 (Appendix A, data not shown) and that c-Fos and c-Jun accumulated over a time-course EPH334 treatment (Figure 5a).

Another established consequence of oxidative stress signaling is the induction of hypoxia [54,55]. As such, we expected the induction and/or stabilization of redox-sensitive regulators of the hypoxia pathway such as HIF-1α. Indeed, EPH334 induced the temporal accumulation of HIF-1α, which increased ~4-fold at 30 min post-treatment and reached a maximum of ~25-fold at 4 h (Figure 5a). HIF-1α accumulation was consistent with the temporal induction of a gene set of hypoxia responsive genes (e.g., *CA9*, *PFKFB4*, and *VEGF*) in our transcriptome analysis (Figure 3 and Appendix A) and the idea that HIF-1α enhances HIV-1 infection by binding hypoxia-responsive elements (HRE) present at the 5′-LTR [10].

Given that EPH334 reactivated latent HIV-1 proviruses from immortalized and primary models of latency as well as aviremic patient samples (Figure 1, Figure 2, Figure 3 and Figure 4) and triggered ROS to induce redox-responsive TFs (AP-1 and HIF-1α) that can impinge on the proviral genome (Figure 5a), we asked whether EPH334 also induced these TFs in primary resting CD4^+^ T cells. For this, we isolated PBMCs from healthy donors to purify naïve CD4^+^ T cells using negative selection, which were polarized to obtain T_CM_ and then treated with EPH334 or vehicle control (DMSO), or left untreated, and performed Western blots to monitor induction of the redox-sensitive factors. Remarkably, a short amount (2 h) of EPH334 treatment, but not DMSO, triggered robust induction of all three redox-sensitive factors (c-Fos, c-Jun and HIF-1α) (Figure 5b). This data extends our results from immortalized to primary CD4^+^ T cells and indicates that EPH334 induced oxidative stress cell signaling in physiologically relevant models of latency.

Because HIV-1 establishes latent reservoirs not only in T_CM_ but also in other T-cell subsets [56,57], we asked whether EPH334 would also trigger the redox-sensitive TFs in the T-cell subsets. To test this idea, we polarized naïve CD4^+^ T cells into the five major subsets (T_CM_, Treg, Th1, Th2 and Th17), allowed them to reach quiescence, and then treated them with EPH334 or vehicle control, or left them untreated. Unexpectedly, EPH334 (2 h, 2.5 μM), but not DMSO, induced AP-1 (c-Fos, c-Jun) and HIF-1α relative to untreated cells, albeit at distinct levels among the T-cell subsets (Appendix A). C-Fos was expressed at higher levels in T_CM,_ intermediate levels in Treg and Th1, and low-to-undetectable levels in Th2 and Th17. Both c-Jun and HIF-1α were expressed in higher levels in T_CM_, Treg, Th1, and Th17 relative to Th2, which had the lowest levels. These data suggest potentially interesting differences in the expression of relevant latency-reversal TFs among T-cell subsets in response to host cell redox alterations.

To test if the redox-sensitive TFs (AP-1 and HIF-1α) bind the proviral genome to promote latent HIV-1 reactivation in response to EPH334, we performed chromatin immunoprecipitation (ChIP) assays on J-Lat 10.6 cell (10 million) nuclear extracts from untreated, vehicle (DMSO), or EPH334-treated cells for 2 h with factor-specific antibodies. Supporting the notion that the EPH334-induced TFs bind to the HIV-1 LTR, we observed that their occupancy increased (~5-fold for c-Fos, ~4-fold for c-Jun, and ~2.5-fold for HIF-1α) in a statistically significant manner in response to EPH334—but not DMSO—treatment over untreated cells (Figure 5c) thereby providing direct evidence of signaling-regulated TF–HIV-1 proviral genome interactions triggered by EPH334.

Given this finding, we then expected that loss of HIF-1α, c-Fos, and/or c-Jun expression should dampen latent HIV-1 reactivation in response to EPH334 treatment. To test this idea, we silenced TF expression using siRNA-mediated RNAi electroporated into J-Lat 10.6 cells for 3 days, which yielded efficient knockdown (KD) (~80% c-Fos, ~65% c-Jun, and ~70% HIF-1α) (Figure 5d), and then treated cells with EPH334 or vehicle control for 24 h or left them untreated to measure latent HIV-1 reactivation using RT-qPCR. Silencing of each TF reduced latent HIV-1 reactivation in a statistically significant manner (~2-fold decrease upon c-Fos KD, ~2-fold decrease upon c-Jun KD, and ~2.5-fold decrease upon HIF-1α KD) (Figure 5e), thus providing strong genetic evidence that the redox-sensitive TFs are required for EPH334-mediated latent HIV-1 reactivation.

Because AP-1 and HIF-1α can cooperate to activate target gene transcription in response to hypoxia [58], we explored whether reduced expression of one TF alters the occupancy of the second TF to the HIV-1 LTR using ChIP assays to test for TF cooperativity. Notably, HIF-1α silencing diminished (~2-fold) c-Fos and c-Jun occupancy at the LTR (Figure 5f), signifying that AP-1 and HIF-1α cooperate to reactivate latent HIV-1 in response to EPH334-provoked host cell redox alterations.

Collectively, the data indicate that EPH334 triggers ROS to induce redox-sensitive TFs that directly occupy—and cooperatively activate—the HIV-1 proviral genome, thereby promoting its reactivation. Below, we investigate the functional relationship between redox-responsive TFs and kinases that phosphorylate them in the reactivation process.

### 2.6. EPH334 Induces Redox-Sensitive Signaling Kinases and TFs That Reactivate Latent HIV-1 in a ROS-Dependent Manner

The above data suggest that EPH334 triggers ROS to induce the synthesis of the TFs (AP-1 and HIF-1α) that mediate latent HIV-1 activation. AP-1 induction in response to cell-extrinsic stimulation is coordinated with the activation of upstream MAPKs that phosphorylate c-Fos and c-Jun for increased DNA binding and transcriptional potential [59,60,61,62,63]. These findings prompted us to test if EPH334, in addition to inducing AP-1 (Figure 5a), also elicited phosphorylation of c-Fos and c-Jun transactivating domains. EPH334, like T-cell-activating stimuli (PMA and Ionomycin), induced c-Fos and c-Jun phosphorylation in J-Lat 10.6 cells (Figure 6a and Appendix A); however, the pattern of site-specific phosphorylation was different. While PMA/Ionomycin elicited robust c-Fos T232 phosphorylation at 60 min (~2.5-fold) and 120 min (~18-fold) poststimulation, EPH334 induced low levels (~1.8-fold) at 120 min poststimulation. Surprisingly, EPH334 triggered high levels of c-Jun S63 (~14-fold) and T91 (~22-fold) phosphorylation at 120 min poststimulation (Figure 6a and Appendix A); whereas PMA/Ionomycin caused lower c-Jun phosphorylation levels (~2-fold S63 and T91) at 120 min poststimulation.

Since EPH334 induced the expression and site-specific phosphorylation of AP-1 subunits, we then expected upstream activation of cell-signaling MAPKs involved in the transduction of host cell redox alterations into transcriptional outputs. For this, the activation status of the three major families of signaling MAPKs (ERK1/2, JNK, and p38), as judged by phosphorylation of their activation segments (T-loops), were surveyed in an acute (0 to 120 min) time course with EPH334 (Figure 6b). We observed earlier ERK1/2 (15 min) and later p38 (60–120 min) activation without significant changes in the total kinase levels. However, JNK was not activated in this short time frame, indicating that EPH334 acutely activates a select subset of MAPKs and that p38 activation coincides with increased levels of c-Jun phosphorylation at S63 and T91. Unlike EPH334, PMA triggered robust, acute (15–30 min) activation of all three MAPKs (JNK, ERK1/2, and p38) (Appendix A), indicating differences in the activation modes for both LRAs. Supporting this notion, EPH334 only stimulated JNK after 8 h of treatment to potentially sustain the host cell redox program (see Figure 7a below).

Given that EPH334 simultaneously induced ROS (Figure 4) and activated the redox-sensitive MAPKs and TFs that reactivate latent HIV-1 (Figure 6a,b), we asked whether these two events were functionally linked. To test this idea, J-Lat 10.6 cells were preincubated with the ROS scavenger NAC or vehicle control (tissue culture media) for 2 h and then treated with EPH334 (or vehicle control DMSO) for a short time point (15 min) required for the rapid activation of signaling MAPKs potentially involved in AP-1 induction and activation. ROS inhibition with NAC largely diminished the rapid and robust EPH334-induced phosphorylation of ERK1/2 (~5-fold) and p38 (~3-fold) without significant changes in total kinase levels (Figure 6c), and without altering JNK levels and activation status (data not shown), as it was not phosphorylated by acute EPH334 treatment (Figure 6a). The data that NAC blocks ERK1/2 activation imply that ROS generation is coupled to upstream MAPK activation.

Since ERK1/2 is activated by upstream kinase MEK, we assessed MEK activation status and found EPH334 also activated MEK, which is partially ROS-dependent (Figure 6c, ~1.5-fold decreased), potentially indicating an ROS-dependent MEK-ERK1/2 kinase axis induced by EPH334 to reactivate latent HIV-1, although other kinases and pathways may exist, as NAC does not fully prevent EPH334-dependent MEK activation.

ERK1/2 phosphorylates and activates the ELK-1 transcription factor, a master regulator of IEGs [64,65]. To test the idea that EPH334-induced ROS signaling promotes ERK1/2-mediated ELK1-dependent expression of *FOS* and *JUN*, we first measured ELK-1 activation (S383 phosphorylation [64,65]) in response to EPH334 alone and the effect of ROS inhibition by pretreating J-Lat 10.6 cells with NAC. While EPH334 induced ELK-1 site-specific phosphorylation (~8.5-fold over untreated cells), ROS inhibition with NAC decreased ELK-1 phosphorylation (~2.4-fold) without affecting ELK-1 total levels (Figure 6d). Further, ROS inhibition with NAC virtually abolished EPH334-mediated c-Fos induction (~12-fold) and its phosphorylation at T232 and partially reduced c-Jun levels (~2.8-fold) with concomitant reductions in S63 and T91 phosphorylation, which is potentially attributed to diminished TF amounts (Figure 6d). Additionally, ROS inhibition with NAC partially (~2-fold) reduced HIF-1α levels (Figure 6d), potentially linking EPH334-mediated HIF-1α induction with the hypoxic response because of reduced oxygen levels and/or reduced HIF-1α degradation.

To assess whether EPH334-mediated ROS response induces AP-1 and HIF-1α synthesis at the transcriptional and post-transcriptional levels, J-Lat 10.6 cells were treated with EPH334 alone or pretreated with NAC as above. RT-qPCR assays revealed that NAC prevented the accumulation of *FOS* and *JUN* transcripts in response to EPH334 treatment (Figure 6e), suggesting that EPH334-induced ROS facilitates the transcriptional induction of *FOS* and *JUN* through the ERK1/2–ELK-1–AP-1 axis. However, despite that HIF-1α protein is largely induced by EPH334 (Figure 6d), *HIF1A* transcript levels remain unaffected by both EPH334 and NAC, like an ROS nonresponsive target gene (*U6*) (Figure 6e). Thus, unlike c-Fos and c-Jun, which are induced transcriptionally and, in an ROS,-dependent manner, HIF-1α induction is likely through partial, ROS-dependent protein stabilization [66]. Together these observations illuminate the divergent pathways leading to stimulation of the redox-sensitive TFs that activate latent HIV-1 in response to host cell redox alterations.

### 2.7. A Redox-Sensitive MEK–ERK1/2 Axis Facilitates EPH334-Induced Latent HIV-1 Reactivation

The above data suggest that redox-sensitive MEK-ERK1/2 activation mediates EPH334-induced latent HIV-1 reactivation. To functionally test if a MEK-ERK1/2 kinase axis was indeed required, MEK (a key component of the RAS-RAF-MEK-ERK1/2 signaling pathway) was first inhibited with MEK pharmacologic inhibitor PD0325901 (MEKi, which reduces P-ERK1/2 levels) for 1 h in J-Lat 10.6 cells, which were then treated with EPH334 or vehicle control (DMSO) for 8 h (Figure 7). PMA/ionomycin was also included in a separate experiment as positive control known to signal through MEK-ERK1/2 and p38 for AP-1 activation (Appendix A).

MEKi almost completely abolished ERK1/2 phosphorylation in response to EPH334, without affecting total ERK1/2 levels (Figure 7a). However, as expected, MEKi did not prevent the activation of the other two signaling kinases, JNK and p38. Of note, MEKi pretreatment increased basal MEK T-loop phosphorylation (~2.8-fold), potentially attributed to a feedback loop, increasing MEK activity to compensate for MEK inhibition. In addition to virtually eliminating ERK1/2 T-loop phosphorylation, MEKi also reduced both c-Fos and c-Jun synthesis and site-specific phosphorylation, mainly c-Jun S63 and T91, as the canonical c-Fos T232 phosphorylation is not activated by EPH334 (Figure 7b). Consistent with the idea of divergent pathways leading to AP-1 and HIF-1α induction, HIF-1α stabilization by EPH334 remains unaltered when preincubating cells with MEKi.

Having validated MEK-specific inhibition with MEKi, EPH334-induced HIV-1 gene expression was then tested by RT-qPCR assays, which revealed that MEKi partially decreased (~2-fold) EPH334-induced latent HIV-1 reactivation (Figure 7c). This partial decrease is consistent with the idea of combinatorial cell-signaling transcriptional programs acting on the LTR in response to EPH334: an MEK-ERK1/2 axis activating AP-1 in addition to HIF-1α activation because of hypoxia signaling.

If MEK–ERK1/2-dependent oxidative stress sensing is directly involved in the process of EPH334-mediated AP-1 gene expression, we would then expect that MEKi should—like the ROS scavenger NAC—block *FOS* and *JUN* synthesis. In agreement, MEKi largely reduced (~4-fold) *FOS* and *JUN* expression (Figure 7c), which is consistent with the loss of latent HIV-1 reactivation (Figure 7c), and reduced c-Fos (~4.5-fold) and c-Jun (~2-fold) levels (Figure 7b).

Together, EPH334 induces ROS to mediate activation of at least two independent redox-sensitive pathways leading to latent HIV-1 reactivation (Figure 8): first, a MEK-ERK1/2 pathway leading to AP-1 (c-Fos and c-Jun) “transcriptional” induction—perhaps through ELK-1—and AP-1 activation via site-specific phosphorylation (primarily at c-Jun S63 and T91); second, a pathway leading to HIF-1α protein stabilization. In this redox-sensitive program, we propose that EPH334 first induces an “acute response”, in which AP-1 subunits are transcriptionally induced and HIF-1α is post-translationally stabilized, and then triggers a “sustained response” in which post-translationally modified AP-1 (c-Jun S63 and T91 and a lack of canonical c-Fos T232) accumulate, potentially contributing to a positive feedback loop, leading to latent HIV-1 reactivation. Accumulation of phosphorylated AP-1 coincides with the peak of kinase (JNK and p38) activation, potentially forming part of a regulatory circuit.

## 3. Discussion

In this study, we implemented a screen approach and found lead molecules with unique chemistries that reactivate latent HIV-1 through an unprecedented mechanism of action. Surprisingly, the top hit in our screen (EPH334) was found to rapidly induce oxidative stress, which activated a redox-responsive program involved in latent HIV-1 transcription and RNA processing. While the role of viral factors inducing oxidative stress to enhance HIV-1 infection of bystander cells is well-established [9], much less is known about the role of oxidative stress during the transition from latency to reactivation. We propose that transiently altering the host cell redox state can switch HIV-1 proviral fate from latent to reactivated with insignificant consequences to the host.

A hallmark of oxidative stress is the accumulation of ROS [67], which are highly reactive oxygen intermediates (including hydrogen peroxide, hydroxyl radical, singlet oxygen, and superoxide anion, among others) known to react with cellular factors such as proteins, nucleic acids and lipids, resulting in altered homeostasis. EPH334 triggered rapid accumulation of ROS, which activated a redox-responsive program constituted by upstream cell-signaling MAPKs (MEK-ERK1/2 axis) and downstream TFs (AP-1 and HIF-1α) that bind the HIV-1 proviral genome in CD4^+^ T cell models of latency. While HIV-1 transcription can be elicited by redox-dependent NF-κB pathways [68,69], EPH334 primarily activated redox-sensitive TFs of the AP-1 family and HIF-1α. Remarkably, EPH334 also promoted latent HIV-1 reactivation from the primary T_CM_ model of latency and from patient-derived resting CD4^+^ T cells, and NAC blocked latent HIV-1 reactivation from the T_CM_ model of latency. These observations suggest that the major discoveries in cell models of latency are recapitulated in several physiologic systems, including resting CD4^+^ T cells isolated from suppressed patients and primary models of latency.

Mechanistically, EPH334 operates at two independent levels: by transcriptionally inducing the expression of *FOS* and *JUN*, and by post-translationally stabilizing HIF-1α. First, EPH334-mediated ROS production activated MEK, which phosphorylates ERK (T202/Y204), which in turn phosphorylates ELK-1 (S383), the master regulator of the IEGs *FOS* and *JUN* [62]. Upon their expression, c-Fos and c-Jun then assemble a transcriptional competent AP-1 complex that becomes phosphorylated in a site-specific manner [62,70,71]. Surprisingly, while known T-cell agonists primarily induce c-Fos T232 and c-Jun S63 phosphorylation, EPH334 elicited sustained c-Jun S63 and T91 phosphorylation without triggering c-Fos T232 phosphorylation, perhaps promoting the formation of a specialized AP-1 dimer to cope with a specific cellular response (redox alterations). Second, EPH334-mediated ROS production also stabilized HIF-1α potentially through the generation of a hypoxic-like state because of the reduction in oxygen levels. While the events of AP-1 and HIF-1α activation appear to occur independently, they are functionally linked as both TFs converge on the latent HIV-1 provirus to promote its activation in a cooperative manner [8,72,73].

Our proposed model is consistent with previous studies reporting that AP-1 binding enhanced HIF-1α transcriptional activation of *VEGF* in other systems (e.g., glioma cells) [74], suggesting AP-1/HIF-1α cooperativity for target gene activation in response to hypoxic stimulation. Notably, activation of ROS and MAPK signaling are coupled: EPH334 first activates an ROS response, which then induces MAPK activation. Supporting our model, pharmacologic inhibition of ROS generation and MAPK signaling (MEK-ERK1/2 axis), as well as silencing of the three master TFs (c-Fos, c-Jun, and HIF-1α), significantly impaired latent HIV-1 reactivation in response to EPH334. Consistently, silencing of the three TFs significantly decreased their binding to the HIV-1 LTR in response to EPH334 treatment in CD4^+^ T-cell-based models of latency, signifying that AP-1/HIF-1α mediates reactivation of latent HIV-1 through direct proviral genome binding.

Given the transcription and latency reactivation levels, EPH334 primarily promotes the host phase of the transcriptional regulatory circuit and does not induce the viral phase, which is required to establish the Tat positive feedback loop for efficient latent HIV-1 reactivation [12]. This idea is consistent with recent discoveries suggesting that another SMORE (JIB-04) destroys the Tat protein [75]. This modus operandi distinguishes EPH334 from other potent LRAs such as TNF and PKC agonists, which normally induce the two phases (host and viral) of the HIV-1 transcriptional program. Because EPH334 does not strongly induce Tat, it may not be potent enough to cause cell death in HIV-1 reservoirs, a desirable effect of LRAs. However, EPH334 showed similar properties in terms of potency when combined with epigenetic regulators (JQ1), suggesting that it can work combinatorially with other LRAs to increase latency-reversal potency, at levels comparable to TCR activation.

Cytokine induction by LRAs is problematic for in vivo administration due to cell damage and lethal inflammatory responses. Besides ROS, the transcriptome analysis revealed that EPH334 induces T-cell activation markers (CD69 and CD25), albeit at much lower levels than strong T-cell activators such as PMA. Further, ROS responses are accompanied by DNA damage, a potential unwanted side effect by this small molecule and any agent working through the ROS pathway. As such, considerations should be given to minimize negative outcomes by performing transient rather than sustained treatments for eradication approaches through ROS pathway manipulation.

Our proposal that rewiring the oxidative stress response in immune cells promotes the latency-reactivation switch through redox-sensitive programs is consistent with two recent lines of research: First, the implication of stress kinases in the acute and broad reactivation of latent HIV-1 in response to a plethora of stimuli (including CD3/CD28, PMA/ionomycin, Prostatin, and SAHA), which yielded increased phosphorylation of IκBα, ERK1/2, p38, and JNK in HIV-1–infected cells across two in vitro latency models [20]. Second, the observed decreases in oxidative stress (antioxidant response) and iron content that accompanied the development of HIV-1 latency [76].

Moreover, recent studies illuminated that latent HIV-1 reactivation from primary CD4^+^ T-cell models of latency is mediated by PKC-independent signaling pathways including two complementary signaling arms of the TCR cascade (RasGRP1-Ras-Raf-MEK-ERK1/2 and PI3K-mTORC2-AKT-mTORC1) [77]. Interestingly, the ERK1/2 kinases were involved in the biogenesis of the P-TEFb kinase through *CCNT1* (cyclin T1 subunit) transcription and CDK9 kinase subunit T-loop site-specific phosphorylation. P-TEFb T-loop phosphorylation is critical for its release from the inhibitory 7SK snRNP complex [78,79] to activate the kinase. Although we have not yet formally tested whether SMOREs promote the release of P-TEFb from the 7SK snRNP complex, we expect this to be the case because the activation of signaling responsive TFs is coordinated with P-TEFb activation for transcriptional stimulation.

Our discoveries spur interesting ideas for future research. First, while the existence of host target factors cannot be simply ruled out, we argue that EPH334 rapidly metabolizes into reactive radicals, eliciting a redox response (Figure 4a). Characterizing the nature of EPH334 metabolization will require chemical approaches beyond the scope of this study. Second, it remains unknown how and where in the cell EPH334 gets metabolized and what specific ROS or highly reactive intermediate(s) trigger the redox-sensing program that reactivates latent HIV-1. Interestingly, drug signature analysis with “Drug Perturbations from GEO 2014” suggested that EPH334 and doxorubicin induce similar transcriptional responses (Appendix A). Doxorubicin reduction by mitochondrial reductases generates anthracycline semiquinone free radicals [80], which are unstable molecules reducing molecular oxygen to superoxide and hydrogen peroxide under aerobic conditions. Moreover, interactions between doxorubicin and iron III generate an iron-based free radical capable of reducing molecular oxygen [80]. Thus, EPH334 may undergo a similar breakdown process that creates a reactive intermediate, triggering the redox-sensing program elucidated in this study. Other hits in the drug signature analysis from DSigDB include known ROS generators such as Ciclopirox and Menadione [81,82,83] (Appendix A).

Third, it is also unclear exactly how ROS induces MAPK activity, whether this occurs through direct oxidative modification and activation of MAPKs (e.g., MEK) and/or by the inactivation or degradation of MAPK phosphatases [84]. Fourth, the kinases that phosphorylate c-Jun at S63 and T91 during the redox-sensing program remain unidentified. While we have observed acute activation of ERK1/2 and p38, both are known Fos kinases [85], and the canonical Jun kinase (JNK) [71] was activated at later time points (8 h post-EPH334 treatment). Fifth, we have not yet uncovered the mechanism by which HIF-1α becomes stabilized with EPH334 treatment, although our data suggest that oxidative stress is partially mediating its stabilization. The precise mechanism may be partly related to the inhibition of the ferrous iron and 2-oxoglutarate (2OG)-dependent HIF prolyl hydroxylases (PHDs), which promote binding of HIF-1α to the Von Hippel Lindau protein, which is a targeting component of an E3 ligase complex, resulting in proteasomal degradation of HIF-1α [86]. During hypoxia (as occurs in EPH334-treated cells) PHD activity is reduced, resulting in the stabilization and accumulation of HIF-1α in the nucleus for target gene transcriptional regulation [86]. Sixth, EPH334 appears to activate other signaling pathways, as revealed by a residual level of latent HIV-1 reactivation in the presence of NAC or MAPK inhibition, potentially explaining the partial contribution of ROS to latent HIV-1 reactivation.

Finally, our studies illuminate the possible general principle that commonly used LRAs also activate cell-signaling transcriptional circuits to reactivate latent HIV-1 in ROS-dependent and/or -independent manners, consistent with recent reports [20,77]. While LRAs are not thoroughly mechanistically characterized, perhaps they function pleiotropically in canonical and noncanonical manners. For example, HDAC inhibitors are assumed to solely work through the canonical HDAC inhibition mechanism, which does not explain how latent proviruses are reactivated, as HDAC inhibition alone does not provide the activating component (e.g., recruitment of Histone Acetyl Transferases). Notably, treatment of J-Lat 10.6 cells with SAHA also elicits ROS and induces IEGs (Appendix A), which are potentially activated through cell-signaling cascades and having a direct participation on latent HIV-1 reactivation. Given the complexity of ROS sensing and signaling, future studies will directly compare the mechanisms underlying ROS generation and latent HIV-1 reactivation by benchmark LRAs and EPH334.

Addressing the above points would certainly hone our understanding on the mechanisms regulating HIV-1 latency and reactivation through host cell redox changes. Our study will guide future efforts to precisely characterize how previously defined and newly discovered small compounds reactivate latent viruses through alteration of host cell biology. Future research will be needed to assess the usefulness of EPH334 in reducing HIV-1 reservoirs in vitro and in vivo for its eradication.

## 4. Methods

### 4.1. SMOREs and EPH334

Formulas and characteristics for all small molecules used in this study will be provided upon request. The structure of EPH334 (MW 315.34 Da, weight 1.4 mg and formula C18H_13_N_5_O) (Figure 9) is provided below.

### 4.2. Cell Culture

Jurkat CD4^+^ T cells (E6.1 clone, ATCC TIB152) and derivatives (Jurkat HIV-1 E4, 10.6, 6.3, 8.4, 15.4, and 9.2 clones, [29,30] (see Appendix A for a complete list of cell lines used in this study) were maintained in RPMI 1640 media (HyClone, Logan, UT, USA, Cat. No. SH30027.FS), supplemented with 10% Fetal Bovine Serum (FBS) (Sigma, St. Louis, MO, USA, Cat. No. F4135) and 1X penicillin/streptomycin (MP Biomedicals, Santa Ana, CA, USA, Cat. No. 1670049) at 37 °C with 5% CO_2_. The E4 clone derives from HIV-1 NL4-3 infectious molecular clone [30] and 10.6, 6.3, 8.4, 15.4, and 9.2 clones derive from the R7/3/GFP molecular clone and contain an *env* frame shift and GFP in place of *nef* (R7/E-/GFP) [29]. The Jurkat clones were treated with the small compounds (Appendix A) at the indicated time points and concentrations. Primary CD4^+^ T cells were isolated and cultured as indicated below.

### 4.3. Nucleofection of Jurkat Cells and RNAi

For nucleofection of J-Lat 10.6 cells with siRNAs for RNAi, cells were maintained in the log phase of growth (~0.5–1 × 10^6^ cells/mL) and passaged every day at a subcultivation ratio of 1:2 for three consecutive days. The day before electroporation, cells were seeded at a density of 1 × 10^6^ cells/mL and incubated overnight. Cells were spun down at 90× *g* for 10 min at room temperature on an Eppendorf Centrifuge 5424, and then the supernatant was removed completely without disturbing the cell pellet. For each nucleofection reaction, 1 × 10^6^ cells were carefully resuspended with 100 μL room-temperature Ingenio Kit solution (Mirus Bio, Cat. No. MIR 50115) and immediately combined the cell suspension with 100 pmoles of siRNA (see Appendix A for a list of all siRNAs used in this study). The mix was then transferred to a 2 mm gap cuvette (Mirus Bio, Cat. No. MIR 50115), placed on a Nucleofector 2b device (Lonza, Cat. No. AAB-1001) and nucleofected using the X-005 program for high-transfection efficiency. The cuvette was then taken out of the holder and the sample was left at room temperature for 10 min. A total of 0.5 mL of preheated, complete RPMI-1640 medium with 10% FBS and 1% penicillin/streptomycin was immediately added to transfer cells from the cuvette to a 12-well plate containing 1 mL preheated complete culture medium (final volume of 1.5 mL). Cells were then incubated in a humidified 37 °C/5% CO_2_ incubator for 72 h to achieve >70% RNAi efficiency and then treated for 8 h with EPH334, vehicle (DMSO) or left untreated. When treatment was complete, cells were divided into two cell pellets of 200,000 cells for Western blot with the indicated antibodies in Figure 5 and 500,000 cells for RNA isolation and RT-qPCR with the indicated oligonucleotides (Appendix A).

### 4.4. Flow Cytometry Analysis

A total of 5 × 10^5^ cells per sample were transferred to an uncoated V-bottom 96-well plate (Nunc, Cat. No. P7241). The samples were spun down at 750× *g* for 5 min at room temperature and washed with 1X PBS (HyClone, Cat. No. SH30028.02). Washed cells were spun down again, and the 1X PBS was aspirated. Cells were fixed using 20 μL of 1% paraformaldehyde (PFA) (Sigma, Cat. No. P6148-500G) at room temperature for 10 min. The PFA was washed with 100 μL of PBS and spun down, buffer was aspirated, and cell pellets were resuspended in 100 μL of 1X PBS. Cells were stained with the indicated antibodies for 30 min at room temperature and then washed twice with 100 μL of 1X PBS. A Stratedigm A600 HTAS 96-well plate reader was used to run the samples. CellCapTure (Stratedigm) was used to visualize the running samples. A set of 20,000 cells were analyzed per sample.

### 4.5. RNA Extraction and RT-qPCR Assays

Isolation of total RNA was performed using the Quick-RNA miniprep kit (Zymo, Irvine, CA, USA, Cat. No. R1055). RNA quality was assessed by running samples on an 1% agarose gel-TBE 0.5X. First strand cDNA synthesis was performed using M-MuLV Reverse Transcriptase with oligo(dT)_18_ and random decamers. Quantitative PCR was performed with a SybrGreen master mix (Applied Biosystems, Waltham, MA, USA, Cat. No. 4309155) on a QuantStudio 3 instrument (Applied Biosystems). Ct values were directly obtained from the instrument. The fold change of the target mRNA levels relative to control was calculated as 2^−ΔΔCt^. A list of DNA oligonucleotides used in RT-qPCR assays can be found in Appendix A.

### 4.6. RNA-seq

Sixteen hours prior to the experiment, cells were seeded at a density of 7.5 × 10^5^ cells/mL in a 6-well plate. After overnight incubation, cells were treated with 2.5 μM EPH334 (1, 8, and 16 h) or 1 μM of SAHA (2, 4, and 8 h) in triplicate. Cell pellets were then collected and stored at −80 °C. Total RNA was isolated using the Quick-RNA Miniprep Kit (Zymo, Cat. No. R1055) according to manufacturer’s instructions. RNA quality was determined with Agilent Tapestation ScreenTape (Agilent, Santa Clara, CA, USA, Cat. No. 5067-5576). An amount of 1 µg of total RNA with ERCC Spike-ins (ThermoFisher, Waltham, MA, USA, Cat. No. 4456740) was used to prepare libraries with KAPA Stranded RNA-seq kit with RiboErase (HMR) (KAPA Biosystems, Wilmington, MA, USA, Cat. No. KK8483) according to manufacturer’s instructions. After final cleanup, libraries were run on Agilent Tapestation DNA D1000 ScreenTape (Agilent, Cat. No. 5067-5582) to confirm fragment size and quantified using a Qubit 2.0 fluorometer. Libraries were submitted to the UT Southwestern McDermott Center Sequencing Core and 75 bp paired-end (PE) reads were generated for both libraries.

### 4.7. RNA-seq Data Analysis

RNA-seq analysis was performed on an Intel E5-2680v3 CPU compute node (with 256 GB RAM running Red Hat Enterprise Linux 7.4) in the high-performance computing (HPC) cluster at UT Southwestern Medical Center. The 75 bp PE RNA-seq reads were trimmed with Trimgalore [87] 0.4.1 using Illumina adapter, PE, and short-read removal options (--illumina --paired --length 30). Trimmed PE reads were aligned to GRCh38 reference genome with HISAT2 [88] 2.1.0 using the multicore (-p 8) option and reporting up to 10 multimappers (-k 10) in SAM format (-S sam_out). Alignments in SAM format were converted to BAM format with Samtools 0.1.19 [89]. Then, the sequences were sorted and duplicates were removed by Picard version 2.18.0 accessed on 25 July 2019 (https://broadinstitute.github.io/picard/). RNA abundance of transcripts found in the GRCh38 reference annotation (gencode.v28.annotation.gtf) was calculated with StringTie [88] v1.3.2d using the multicore (-p 36) and first-stranded library (--rf) options. RNA abundance was reported in Ballgown-compatible format (-B) and only for transcripts found in the annotation file (-e). RNA abundance was converted to read counts with prepDE.py using a length of 75 (-l 75). DEGs were identified with DESeq2 (version 1.24.0) [90] using an FDR cutoff value < 0.05 and MA plots were plotted with R scripts using ggpubr (0.2.2).

### 4.8. Oxidative Stress Detection

Sixteen hours prior to experiment, cells were seeded at 7.5 × 10^5^ cells/mL in a 48-well plate at 500 μL/well. After 16 h incubation, cells were treated with the specified compounds (DMSO, EPH334 or Menadione as positive control). During the final 30 min of treatment with experimental compounds, CellROX Green reagent (Thermo Fisher, Cat. No. C10444) was added at 5 μM final concentration and returned to the incubator. Once the desired treatment time was reached, cells were washed 3 times with 1X PBS, then fixed with 2% PFA solution at 4 °C for 15 min. After fixation, cells were washed 3 times with 1X PBS and resuspended in 1X PBS containing 0.5% FBS and 1 mM EDTA at a density of 2.5 × 10^6^ cells/mL. Cells were analyzed immediately on a Stratedigm A600 HTAS 96-well plate reader with instrument settings consistent with GFP detection. Data analysis was performed with FlowJo version 10.1 (TreeStar Inc., San Francisco, CA, USA).

### 4.9. ChIP-qPCR Assays

ChIP assays in J-Lat 10.6 cells were performed as previously described [91]. Purified cell nuclei (10 million cells) were sonicated 60 cycles (30 s on/30 s off) on a Bioruptor UCD-300 water bath (Diagenode) to obtain DNA fragments of an average size of ∼300 bp. Antibodies were conjugated to 50 μL of 50% slurry Protein G Dynabeads at 4 °C for 2 h and added to purified sonicated cell nuclei (see Appendix A for complete list of antibodies and dilutions used). ChIP assays were performed with protein extracts from the indicated cells and using the antibodies indicated followed by qPCR with the indicated amplicons. The ChIP-qPCR data were normalized using the “Percent Input Method”, which includes normalization for background and input chromatin used for each ChIP. ChIP signals were divided by signals obtained from the input sample (1% of starting chromatin), which signifies the amount of chromatin used per ChIP. Values represent the percentage (%) of input DNA immunoprecipitated (IP DNA) presented after background (normal IgG) subtraction and are the average of three independent experiments.

### 4.10. Isolation of Naïve CD4^+^ T Cells and Generation of T_CM_

Isolation of naïve CD4^+^ T cells and T_CM_ generation were performed as recently described [92]. Briefly, PBMCs were isolated from 50 mL leukopaks of healthy, random, and deidentified donor blood samples (Gulf Coast Regional Blood Center, Houston, TX), in accordance with local ethical standards, using Lymphoprep density gradient (STEMCELL, Cat. No. 07801). Naïve CD4^+^ T cells were isolated from PBMCs following the instruction of EasySep™ Human Naïve CD4^+^ T Cell Isolation Kit (STEMCELL, Cat. No. 558521). Purity was analyzed by flow cytometry comparing the presence of well-defined markers to isotype controls: CD4-PerCP-Cy5.5 (eBioscience, San Diego, CA, USA, Cat. No. 45-0049-42) 0.125 μg/test, control: mouse IgG1k isotype PerCP-Cy5.5 (eBioscience, Cat. No. 45-4714-82) 0.125 μg /test; CD3-PE/Cy7 (BioLegend, San Diego, CA, USA, Cat. No. 300419) 0.125 μg/test, control: mouse IgG1k isotype PE/Cy7 (Invitrogen, Waltham, MA, USA, Cat. No. 25-4714-80) 0.125 μg/test; CD45RO-PE (eBioscience, Cat. No. 12-0457-41) 0.125 μg/test, control: mouse IgG2a isotype PE (eBioscience, Cat. No. 12-4724-42) 0.125 μg/test; CD45RA-FITC (eBioscience, Cat. No. 11-0458-42) 0.125 μg/test; CD25-FITC (eBioscience, Cat. No. 11-0257-42) 0.125 μg/test, control: mouse IgG2bκ isotype FITC (eBioscience, Cat. No. 11-4732-4111-4732-41) 0.125 μg/test. Cells were then activated by culturing with 1 μg/mL anti-IL-4 (PeproTech, Cat. No. 500-P24), 2 μg/mL anti-IL-12 (PeproTech, London, UK, Cat. No. P154G), 10 ng/mL TGF-β1 (PeproTech, Cat. No. 100-21) and Dynabeads human T-activator anti-TCR/anti-CD28 (Gibco, Grand Island, NE, USA, Cat. No. 11132) (1 bead/cell). After 3 days, beads were removed by a column-free magnetic separation. Cells were cultured in complete media with 30 IU/mL IL-2 (Roche, Basel, Switzerland, Cat. No. 202-IL). Media containing IL-2 were replenished daily for 10 days, followed by every other day, and cells were maintained at 1 × 10^6^/mL. Transition from effector into central memory (T_CM_) was monitored by flow cytometry analysis of intracellular Ki67 (eBioscience, Cat. No. 12-5699-41) 0.125 μg/test, control: mouse IgG1κ Isotype PE (eBioscience, 12-4714-41) 0.125 μg/test and surface CD45RA-FITC/CD45RO-PE, in where low expression of Ki67 (<5%) was considered low-dividing, and therefore evidence of quiescence.

### 4.11. Primary Model of Latency

VSV-G pseudotyped viruses (pNL4.3-delta*Env*-nLuc-2A*Nef*-VSVG) were produced by cotransfecting pNL4.3-delta*Env*-nLuc-2A*Nef* and pCMV-VSVG into 293T cells. Cell supernatants were collected and filtered with a 0.22 μm filter after 2 days. Viruses were aliquoted and stored at −80 °C. T_CM_ were generated as explained in the above section. At days 3 or 4 postisolation, cells were infected with pseudotyped viruses (pNL4.3-delta*Env*-nLuc-2A*Nef*-VSVG) using spinoculation method. After 2 days, HIV-1-infected cells were analyzed by flow cytometry by staining with Fixable Viability Dye eFluor 450 (eBioscience, Cat. No. 65-0863-14) and anti-CD4-APC (eBioscience, Cat. No. 17-0149-41) for 30 min. After washing with 1X PBS containing 3% FBS, cells were fixed and permeabilized using Cytofix/Cytoperm (BD, East Rutherford, NJ, USA, Cat. No. 554714) for 30 min and then stained with anti-p24-FITC (KC57, Beckman Coulter, Pasadena, CA, USA, Cat. No. 6604665). At day 17, CD4^+^ T cells were isolated using Dynabeads CD4^+^ T-cell positive isolation Kit. At day 18, cells were seeded into 96-well plates, pretreated for 2 h with NAC (10 mM) or vehicle control (tissue culture media) and then treated with EPH334 (2.5 μM) or vehicle control (DMSO) for 24 h. Luciferase of supernatant (Nano-Luc) was measured using Nano-Glo Luciferase assay system (Promega, Madison, WI, USA, Cat. No. N1110).

### 4.12. Generation of Polarized Effector CD4^+^ T Cells

To generate Th1, Th2, Th17, and Treg cells, naïve CD4^+^ T cells were isolated from PBMCs as described above. Cells were activated by culturing with Dynabeads human T-activator anti-TCR/anti-CD28 (Gibco, Cat. No. 11132) (1 bead/cell) and cytokines and neutralizing antibodies listed below. After 3 days, polarization was verified by flow cytometry of intracellular or nuclear stained transcriptional factors: Th1: T-bet-eFluor 450 (eBioscience, Cat. No. 48-5825-80), Th2: GATA3-Alexa Fluor 488 (eBioscience, Cat. No. 53-9966-41), Th17: RORγt-PE (Invitrogen, Waltham, MA, USA, Cat. No. 12-6988-80), Treg: FOXP3-FITC (eBioscience, Cat. No. 11-4776-42). Beads were removed by a column-free magnetic separation. Cells were cultured for 7 days in media containing respective cytokines and neutralizing antibody cocktail plus 30 IU/mL IL-2 (R&D Systems, Santa Clara, CA, USA, Cat. No. 202-IL). Cells were counted and adjusted to 1 x 10^6^/mL every other day. Th1: IL-12 (50 U/mL) (R&D, Cat. No. 219-IL-005), and anti-IL-4 (2 μg/mL) (PeproTech, Cat. No. 500-M04). Th2: IL-4 (20 ng/mL) (R&D, Cat. No. 204-IL-010), anti-IL-12 (5 μg/mL) (PeproTech, Cat. No. 500-M12), and anti-IFNγ (5 μg/mL) (PeproTech, Cat. No. 500-M90). Th17: TGF-β (5 ng/mL) (PeproTech, Cat. No. 100-21C), anti-IL-4 (500 ng/mL) (PeproTech, Cat. No. 500-M04), anti-IFNγ (10 ng/mL) (PeproTech, Cat. No. 500-M90), IL-23 (50 ng/mL) (PeproTech, Cat. No. 200-23), IL-6 (30 ng/mL) (PeproTech, Cat. No. 200-06), and IL-1β (10 ng/mL) (PeproTech, Cat. No. 200-01B). Treg: TGF-β (5 ng/mL) (PeproTech, Cat. No. 100-21C), anti-IL-4 (500 ng/mL) (PeproTech, Cat. No. 500-M04), anti-IFNγ (10 ng/mL) (PeproTech, Cat. No. 500-M90), and anti-IL-12 (500 ng/mL) (PeproTech, Cat. No. 500-M12).

### 4.13. Isolation of Resting CD4^+^ T Cells from Patient Samples and REVEAL Assay

PBMCs from aviremic patient samples were isolated from peripheral blood obtained via phlebotomy from aviremic individuals living with HIV-1 infection on antiretroviral therapy according to active IRB protocols 58246 at the University of Utah and STU 082014-028 at The University of Texas Southwestern Medical Center. Resting CD4^+^ T cells were purified using a custom kit (STEMCELL, Vancouver, BC, Canada, Cat. No. 17962). Cell viability was measured with Trypan blue. CD4^+^ T-cell purity was measured through flow cytometry analysis by staining with anti-CD4-APC (eBioscience, Cat. No. 17-0149-41). CD4^+^ T-cell activation was measured through flow cytometry analysis by staining with anti-CD25-FITC (eBioscience, Cat. No. 11-0257-42) and anti-CD69-PE (eBioscience, Cat. No. 12-0699-42). Ex vivo LRA treatments of patient resting CD4^+^ T cells and quantitation of secreted HIV-1 RNAs present in cell-associated RNA was performed using the REVEAL assay with the same 3′-LTR primers employed by Spivak et al. [37]. Briefly, cells were cultured for 24 h in the absence of cytokines prior to isolation of cell-associated RNA in aliquots of 5 million cells per each experimental and control condition. RT-qPCR readouts of each condition were performed in triplicate, and the results represent the median value of three independent measurements. As shown in Figure 2i, the absolute values and fold changes increased after exposure to EPH334, but statistical significance was not computed, likely due to small n because of the constrained number of resting CD4^+^ T cells obtained from the various participants. The limit of detection (LOD) for this experiment (20 copies/million cell equivalents) was evaluated by quantifying copies of cell-associated HIV-1 mRNA.

### 4.14. Data Deposition

RNA-seq raw and processed sequencing data of EPH334 and SAHA treated J-Lat 10.6 cells have been submitted to NCBI Gene Expression Omnibus under accession number GSE144552: https://www.ncbi.nlm.nih.gov/geo/query/acc.cgi?acc=GSE144552 (Public on 30 July 2020).

## Figures and Tables

**Figure 1 viruses-14-02288-f001:**
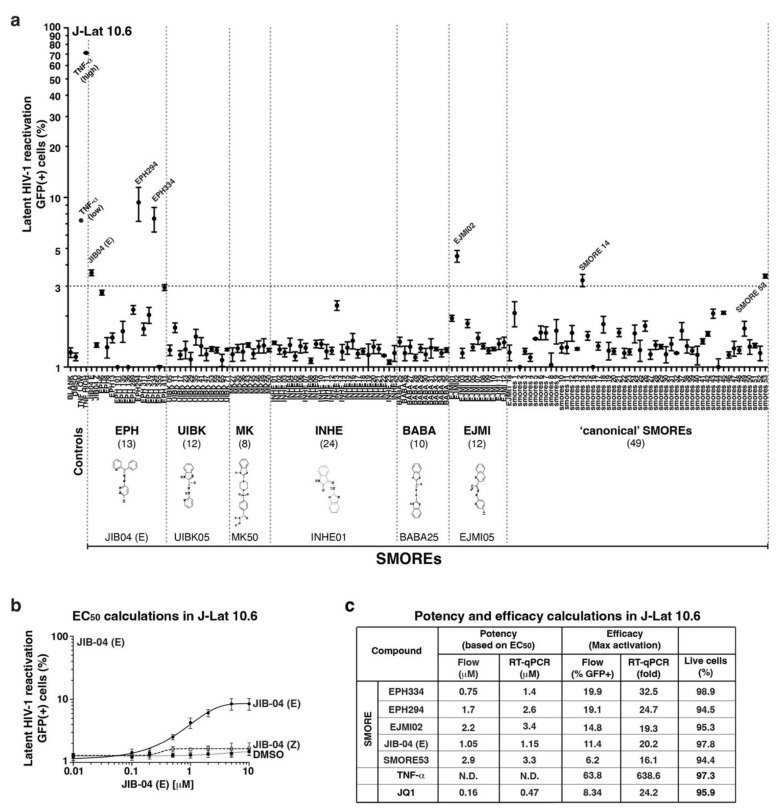
Small compound targeted screen data for latent HIV-1 reactivation, potency, and efficacy studies. (**a**) J-Lat 10.6 cells were treated as indicated in the text and latent HIV-1 reactivation measured by flow cytometry to assess levels of GFP expression. The number of compounds per group is indicated in brackets and representative chemical structures of one compound per group are shown. The dashed line differentiates compounds that activate more than 3-fold over control samples (DMSO) compared to those that do not activate or activate less than 3-fold. TNF-α high (10 ng/mL) or low (1 ng/mL) concentrations were used as positive controls to gauge the dynamic range of the response. The percentage of GFP-positive cells from three independent runs is indicated (mean ± SEM; *n* = 3). (**b**) Example of EC_50_ calculations in J-Lat 10.6 cells by flow cytometry. Plots were derived using a nonlinear regression equation and an asymmetric sigmoidal curve fit to the raw values. GFP background was about 1.5%. The number of GFP positive cells from three independent runs is indicated (mean ± SEM; *n* = 3). (**c**) Potency and efficacy values. J-Lat 10.6 cells were treated with increasing drug concentrations (0.15, 0.3, 0.6, 1.15, 2.5, 5, and 10 μM) and potency and efficacy calculated by flow cytometry and RT-qPCR. Quantitation of percentage of GFP-positive cells by flow cytometry from max dose. Quantitation of HIV-1 RNA levels with VQA amplicon normalized to *RPL19* by RT-qPCR at max dose. The SEM from three independent experiments is indicated. N.D. denotes not determined.

**Figure 2 viruses-14-02288-f002:**
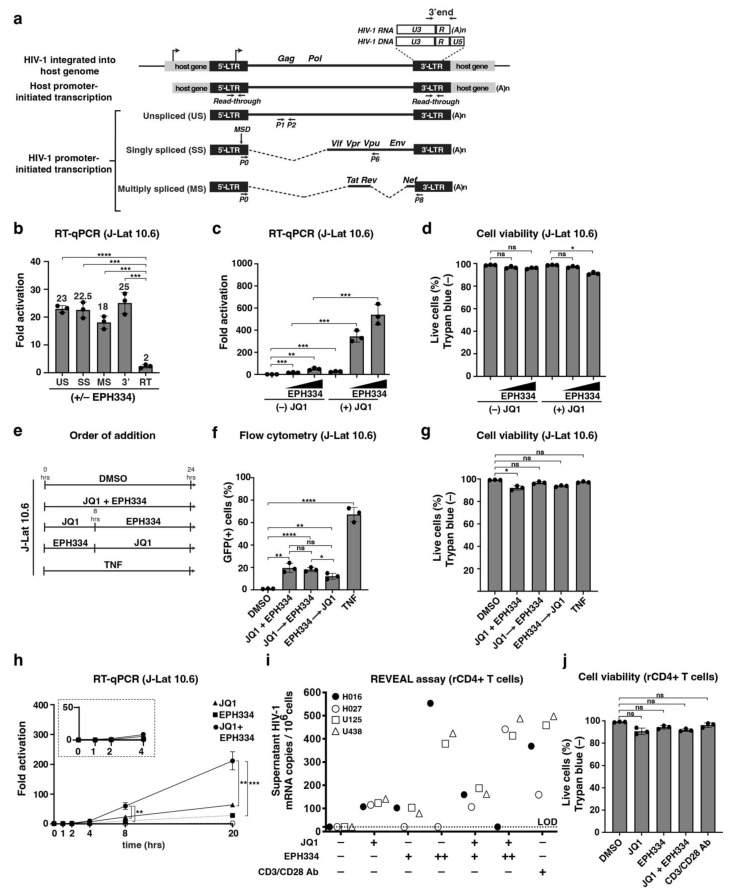
SMOREs induce HIV-1 promoter-initiated transcription, HIV-1 RNA processing, and combinatorial reactivation with bromodomain inhibitors. (**a**) Scheme of HIV-1 RNA transcription, splicing, and 3′-end processing. Using a common multiple splice donor (MSD) present at the 5′-end of the pre-mRNA, HIV-1 RNA is correctly spliced in the singly spliced (SS) and multiply spliced (MS) forms. A primer pair to measure read-through (RT) transcripts was also included. The arrow indicates the transcription start site (TSS) within the 5′-LTR. (**b**) SMOREs induce normal HIV-1 RNA splicing and 3′-end processing. Data are representative of one SMORE (EPH334). J-Lat 10.6 cells were treated with EPH334 (1 μM) for 24 h and RNA isolated. Quantitation of HIV-1 RNAs was done with RT-qPCR assays with the indicated primers, normalized to *RPL19* (mean ± SEM; *n* = 3) and expressed as “Fold activation” +EPH334 over untreated cells. TNF-α and JQ1 were tested in parallel, and datasets were obtained as positive induction controls (data not shown). *p*-values (* *p* < 0.05, ** *p* < 0.01, *** *p* < 0.001, **** *p* < 0.0001) were determined by unpaired Student’s t test. *p* < 0.05 was considered statistically significant. (**c**) Reactivation data of J-Lat 10.6 cells treated with DMSO, increasing concentrations (0.5 and 1 μM) of SMORE (EPH334) alone or in combination with JQ1 (0.5 μM) for 24 h. Quantitation of HIV-1 RNAs (VQA amplicon) normalized to *RPL19* by RT-qPCR (mean ± SEM; n = 3). *p*-values (* *p* < 0.05, ** *p* < 0.01, *** *p* < 0.001, **** *p* < 0.0001) were determined by unpaired Student’s *t* test. *p* < 0.05 was considered statistically significant. (**d**) Percentage of live cells in samples of panel (**c**) measured by incorporation of Trypan Blue from three independent runs is indicated (mean ± SEM; *n* = 3). *p*-values (* *p* < 0.05, ** *p* < 0.01, *** *p* < 0.001, **** *p* < 0.0001, ns = not significant) were determined by unpaired Student’s *t* test. *p* < 0.05 was considered statistically significant. (**e**) Experimental scheme in which J-Lat 10.6 cells were treated simultaneously or sequentially with JQ1 and EPH334 and without wash out, in addition to DMSO (vehicle control) and TNF-α (used as positive control). (**f**) Reactivation data (from experimental design shown in panel (**e**)) measured by flow cytometry. The percentage of GFP positive cells from three independent runs is indicated (mean ± SEM; *n* = 3). *p*-values (* *p* < 0.05, ** *p* < 0.01, *** *p* < 0.001, **** *p* < 0.0001, ns = not significant) were determined by unpaired Student’s *t* test. *p* < 0.05 was considered statistically significant. (**g**) Percentage of live cells in samples of panel (**f**) measured by incorporation of Trypan Blue from three independent runs is indicated (mean ± SEM; *n* = 3). *p*-values (* *p* < 0.05, ** *p* < 0.01, *** *p* < 0.001, **** *p* < 0.0001, ns = not significant) were determined by unpaired Student’s t test. *p* < 0.05 was considered statistically significant. (**h**) EPH334 and EPH334+JQ1 drug combination activate HIV-1 RNA synthesis with rapid kinetics. J-Lat 10.6 cells were treated with EPH334, JQ1, EPH334+JQ1, and TNF-α (10 ng/mL, positive control, data not shown), and kinetics of latent HIV-1 reactivation were measured by RT-qPCR assays. The fold activation for each time point over the zero time point is plotted. Quantitation of HIV-1 RNAs (3′-end, VQA amplicon) normalized to *RPL19* by RT-qPCR (mean ± SEM; *n* = 3). *p*-values (* *p* < 0.05, ** *p* < 0.01, *** *p* < 0.001, **** *p* < 0.0001, ns = not significant) were determined by unpaired Student’s *t* test. *p* < 0.05 was considered statistically significant. The inset shows an expansion of the data between 0–4 h treatments. (**i**) Aviremic patient rCD4^+^ T cell purification and quantitation of released virion RNA in response to the indicated compound treatments: JQ1 (0.5 μM), low (+) EPH334 (1 μM) and high (++) EPH334 (10 μM). HIV-1 RNA (copies/mL culture medium) were quantified in the culture supernatant using TaqMan PCR by interpolating on a HIV-1 copy number standard curve in which increasing copies of input VQ plasmids (pVQA) [33] were used to calculate the threshold cycle number (Ct) of amplification. (**j**) Percentage of live cells in samples of panel (**i**) measured by incorporation of Trypan Blue from three independent runs is indicated (mean ± SEM; *n* = 3). Data from one representative donor out of 4 donors are shown. For the combination JQ1 + EPH334 only the max concentration of EPH334 (10 μM) was tested. *p*-values (* *p* < 0.05, ** *p* < 0.01, *** *p* < 0.001, **** *p* < 0.0001, ns = not significant) were determined by unpaired Student’s *t* test. *p* < 0.05 was considered statistically significant.

**Figure 3 viruses-14-02288-f003:**
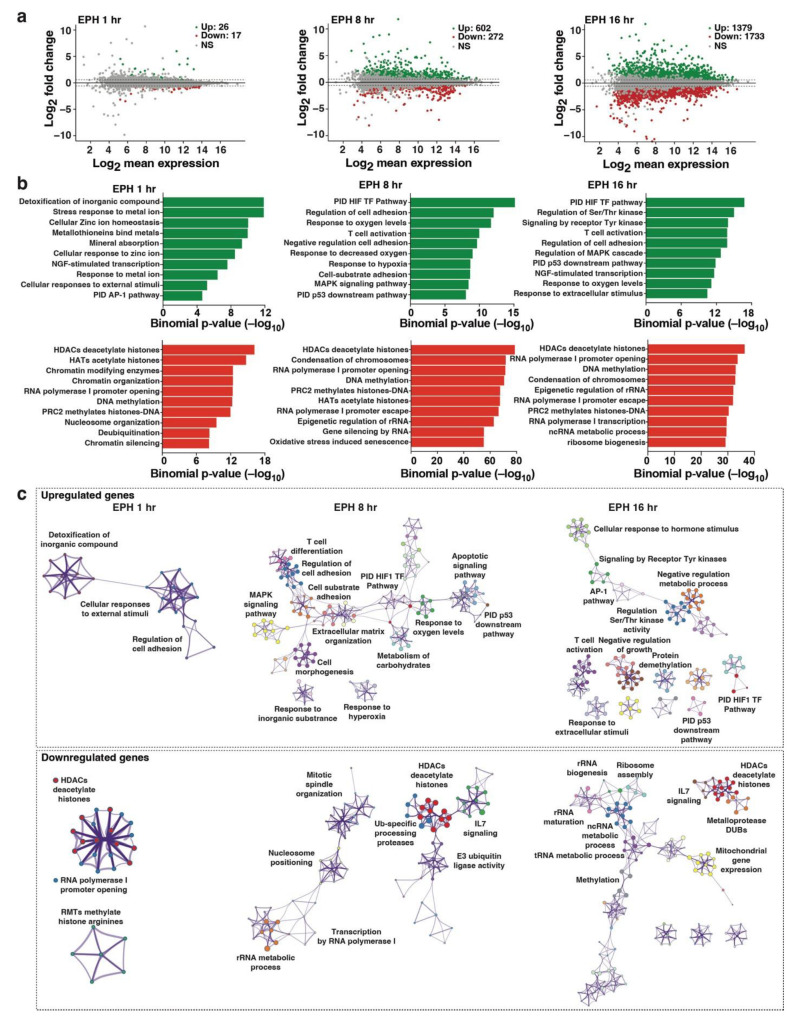
Transcriptional profiling reveals rapid activation of oxidative stress and hypoxia, leading to activation of master transcriptional regulators. (**a**) Scatterplots of the differentially expressed genes identified by RNA-seq of J-Lat 10.6 cells treated with EPH334 (2.5 μM) for the indicated time points (*n* = 3, false discovery rate [FDR] < 0.05). The number of upregulated (Up), downregulated (Down), and not statistically significant (NS) genes is indicated in each plot. (**b**) Functional annotation of biological processes enriched at upregulated and downregulated genes for the indicated time points. (**c**) Metascape enrichment network visualization showing the intracluster and intercluster similarities of enriched terms for upregulated and downregulated genes. Enrichment networks were created by representing each enriched term (enriched biological process) as a node (circle) and connecting pairs of nodes (lines) based on differential gene enrichment. Cluster annotations are color-coded for easier visualization.

**Figure 4 viruses-14-02288-f004:**
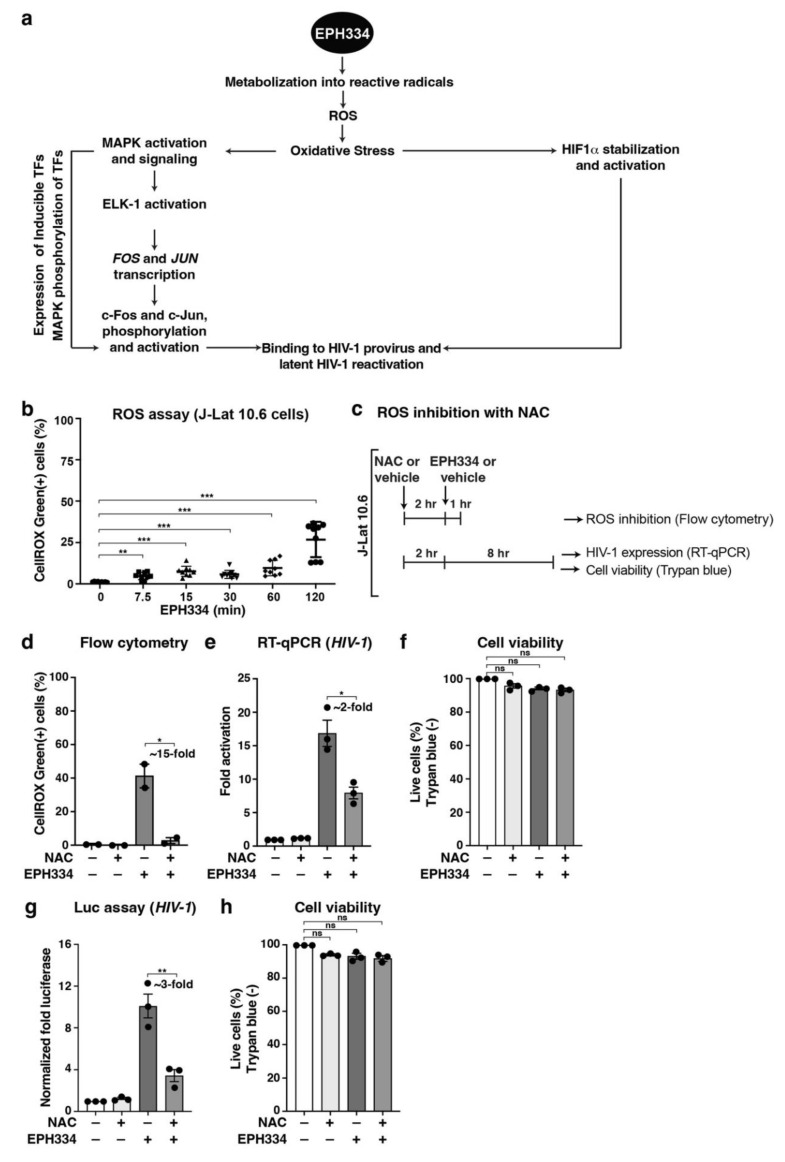
EPH334 triggers oxidative stress to reactivate latent HIV-1. (**a**) Transcriptome-based model predicting EPH334 elicits oxidative stress, thereby inducing several redox-responsive programs that impinge on the latent HIV-1 provirus. (**b**) Measurement of oxidative stress in J-Lat 10.6 cells treated with EPH334 (2.5 μM) for the indicated time points as judged based on the percentage of CellROX Green Dye-positive cells determined by flow cytometry. *p*-values (* *p* < 0.05, ** *p* < 0.01, *** *p* < 0.001, ns = not significant) were determined by unpaired Student’s *t* test. *p* < 0.05 was considered statistically significant. (**c**) Experimental scheme in which J-Lat 10.6 cells were pretreated for 2 h with NAC (10 mM) or vehicle control (tissue culture media) and then treated with EPH334 (2.5 μM) or vehicle control (DMSO) for the indicated time points and then used in the described assays. (**d**) J-Lat 10.6 cells were treated with NAC, EPH334, or vehicle controls, and the percentage of CellROX Green Dye-positive cells was measured by flow cytometry. The percentage of GFP-positive cells from two independent runs is indicated (mean ± SEM; *n* = 2). *p*-values (* *p* < 0.05, ** *p* < 0.01, *** *p* < 0.001, ns = not significant) were determined by unpaired Student’s *t* test. *p* < 0.05 was considered statistically significant. (**e**) Reactivation data of J-Lat 10.6 cells treated with NAC, EPH334, or vehicle controls for 24 h. Quantitation of HIV-1 RNAs (3′-end VQA amplicon) normalized to *RPL19* by RT-qPCR (mean ± SEM; *n* = 3). *p*-values (* *p* < 0.05, ** *p* < 0.01, *** *p* < 0.001, ns = not significant) were determined by unpaired Student’s *t* test. *p* < 0.05 was considered statistically significant. (**f**) Percentage of live cells in samples of panel (**e**) measured by incorporation of Trypan Blue from three independent runs is indicated (mean ± SEM; *n* = 3). *p*-values (* *p* < 0.05, ** *p* < 0.01, *** *p* < 0.001, ns = not significant) were determined by unpaired Student’s *t* test. *p* < 0.05 was considered statistically significant. (**g**) Reactivation data of primary model of latency (T_CM_) pretreated with NAC (10 mM) or vehicle control and then treated with EPH334 (2.5 μM) or vehicle control (DMSO) for 24 h as in panel (**c**). Quantitation of Luciferase production off the reporter virus used to generate the primary model of latency normalized to total protein content from BCA assays is shown (mean ± SEM; *n* = 3). *p*-values (* *p* < 0.05, ** *p* < 0.01, *** *p* < 0.001, ns = not significant) were determined by unpaired Student’s t test. *p* < 0.05 was considered statistically significant. (**h**) Percentage of live cells in samples of panel (**g**) measured by incorporation of Trypan Blue from three independent runs is indicated (mean ± SEM; *n* = 3). *p*-values (**p* < 0.05, ** *p* < 0.01, *** *p* < 0.001, ns = not significant) were determined by unpaired Student’s *t* test. *p* < 0.05 was considered statistically significant.

**Figure 5 viruses-14-02288-f005:**
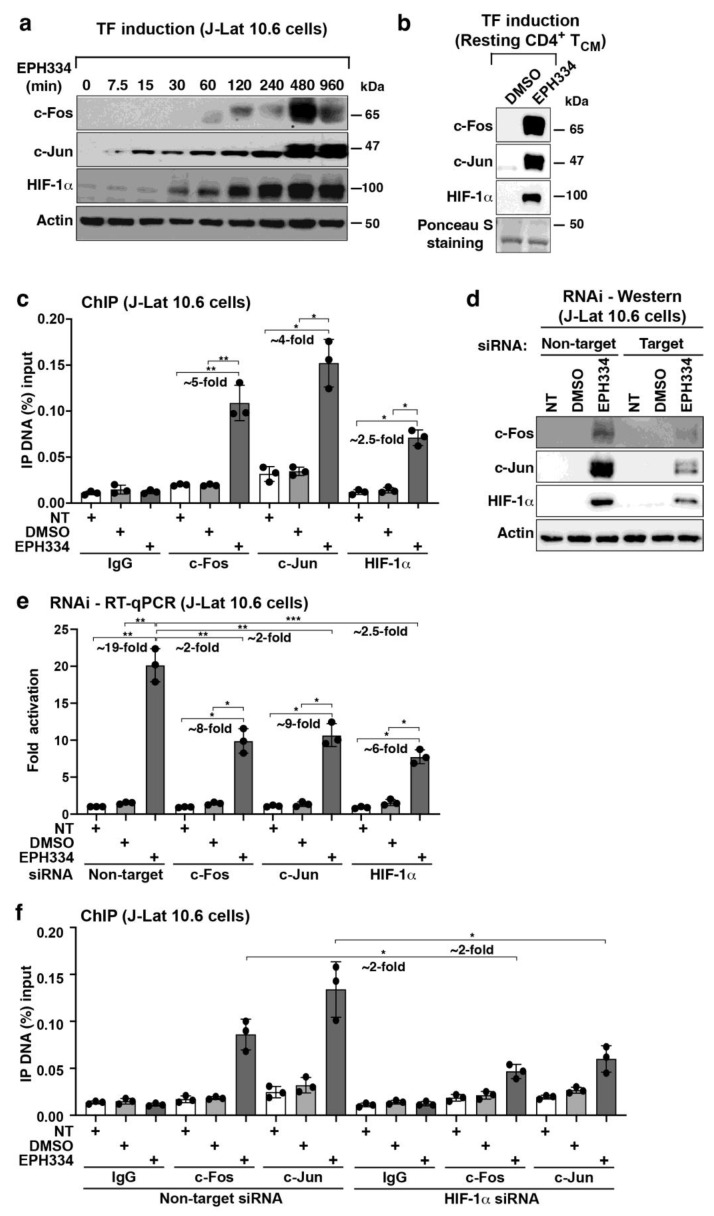
EPH334 induces redox-sensitive transcription factors that directly activate the HIV-1 proviral genome. (**a**) Western blots of J-Lat 10.6 cells showing EPH334-mediated temporal induction of c-Fos, c-Jun, and HIF-1α. (**b**) Western blots of primary resting CD4^+^ T central memory (T_CM_) cells showing EPH334-mediated induction of c-Fos, c-Jun, and HIF-1α. Ponceau S staining was used to measure total protein levels. (**c**) ChIP-qPCR assays of J-Lat 10.6 cells showing EPH334-mediated occupancy of c-Fos, c-Jun, and HIF-1α at the HIV-1 LTR promoter (mean ± SEM, *n* = 3). *p*-values (* *p* < 0.05, ** *p* < 0.01, *** *p* < 0.001, ns = not significant) were determined by unpaired Student’s t test. *P* < 0.05 was considered statistically significant. (**d**) Western blots of J-Lat 10.6 cells treated with the indicated siRNAs and then incubated with EPH334 (2.5 μM) for 24 h or vehicle control, or not treated (NT) as negative control. (**e**) Reactivation data of J-Lat 10.6 cells from panel (**d**). Quantitation of HIV-1 RNAs (3′-end VQA amplicon) normalized to *RPL19* by RT-qPCR (mean ± SEM; *n* = 3). P-values (* *p* < 0.05, ** *p* < 0.01, *** *p* < 0.001, ns = not significant) were determined by unpaired Student’s *t* test. *p* < 0.05 was considered statistically significant. (**f**) ChIP-qPCR assays of J-Lat 10.6 cells from panel (**d**) showing that EPH334 mediates the cooperative binding of AP-1/HIF-1α to the HIV-1 LTR promoter. *p*-values (* *p* < 0.05, ** *p* < 0.01, *** *p* < 0.001, ns = not significant) were determined by unpaired Student’s *t* test. *p* < 0.05 was considered statistically significant.

**Figure 6 viruses-14-02288-f006:**
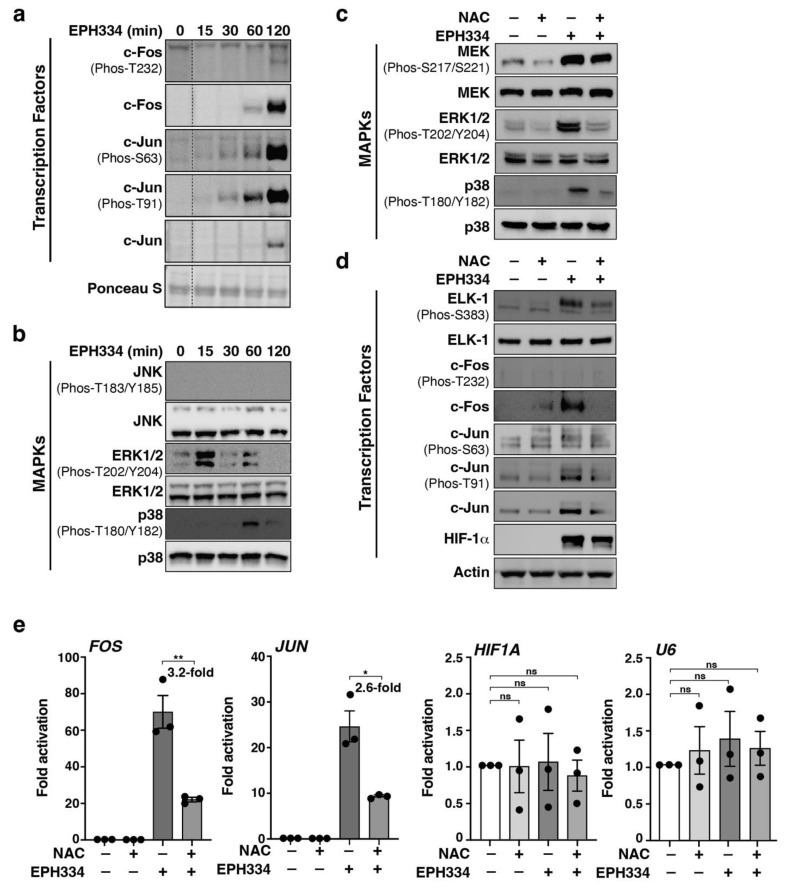
Pharmacologic inhibition of ROS generation blocks EPH334-mediated transcription factor induction and phosphorylation, thereby dampening latent HIV-1 reactivation. (**a**,**b**) Western blots of J-Lat 10.6 cells treated during a time course with EPH334 (2.5 μM) and probed with the indicated antibodies. The dashed line in panel (**a**) indicates that the blot was cropped. Time zero was placed before the treatment. See original complete blot in Appendix A. (**c**,**d**) Western blots of J-Lat 10.6 cells pretreated for 2 h with NAC (10 mM) or vehicle control (tissue culture media) and then treated for 15 min (for the MAPKs) or 8 h (for the transcription factors) with EPH334 (2.5 μM) or vehicle control (DMSO) and then probed with the indicated antibodies. (**e**) RT-qPCR assays of J-Lat 10.6 cells pretreated for 2 h with NAC (10 mM) or vehicle control (tissue culture media) and then treated for 8 h with EPH334 (2.5 μM) or vehicle control (DMSO). Quantitation of the indicated RNAs was normalized to *RPL19* by RT-qPCR (mean ± SEM; *n* = 3). *p*-values (* *p* < 0.05, ** *p* < 0.01, ns = not significant) were determined by unpaired Student’s *t* test. *p* < 0.05 was considered statistically significant.

**Figure 7 viruses-14-02288-f007:**
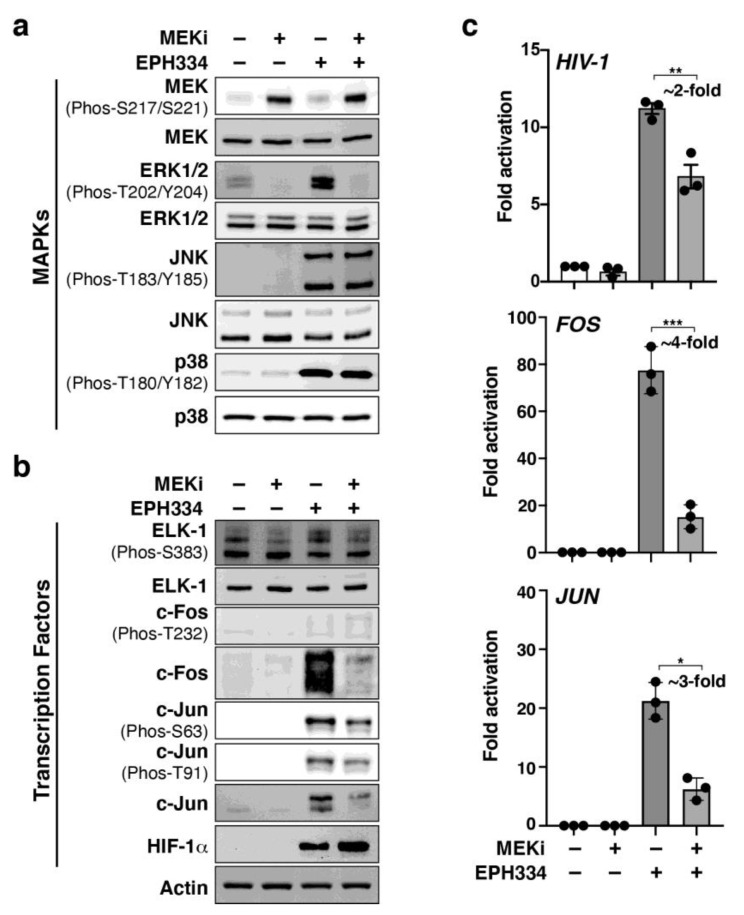
Pharmacologic inhibition of MEK signaling blocks EPH334-mediated AP-1 induction, thereby dampening latent HIV-1 reactivation. (**a**,**b**) Western blots of J-Lat 10.6 cells pretreated for 1 h with MEKi (PD0325901, 1 μM) or vehicle control (DMSO) and then treated for 8 h with EPH334 (2.5 μM) or vehicle control (DMSO) and probed with the indicated antibodies. (**c**) RT-qPCR assays of J-Lat 10.6 cells pretreated for 1 h with MEKi (PD0325901, 1 μM) or vehicle control (DMSO) and then treated for 8 h with EPH334 (2.5 μM) or vehicle control (DMSO). Quantitation of the indicated RNAs normalized to *RPL19* by RT-qPCR (mean ± SEM; *n* = 3). *p*-values (* *p* < 0.05, ** *p* < 0.01, *** *p* < 0.001, ns = not significant) were determined by unpaired Student’s *t* test. *p* < 0.05 was considered statistically significant.

**Figure 8 viruses-14-02288-f008:**
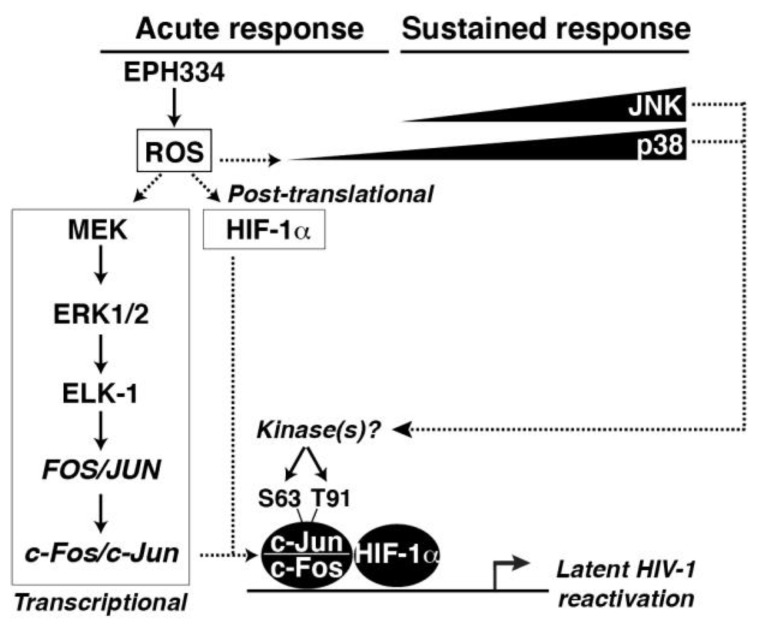
Simplified model of host cell redox alterations influencing HIV-1 proviral transcription and reactivation from latency. First, EPH334 induces an “acute response” by eliciting ROS-dependent induction of the MEK-ERK1/2 axis and p38. The MEK-ERK1/2 axis, probably through ELK-1, transcriptionally induces *FOS* and *JUN* to form the AP-1 dimeric TF complex. ROS-mediated hypoxia simultaneously induces HIF-1α through a non-transcriptional mechanism. Second, EPH334 induces a sustained response through which the levels of the induced TFs and site-specific phosphorylation of c-Jun (S63 and T91) is maintained. Consequently, AP-1 and HIF-1α cooperate to activate latent HIV-1.

**Figure 9 viruses-14-02288-f009:**
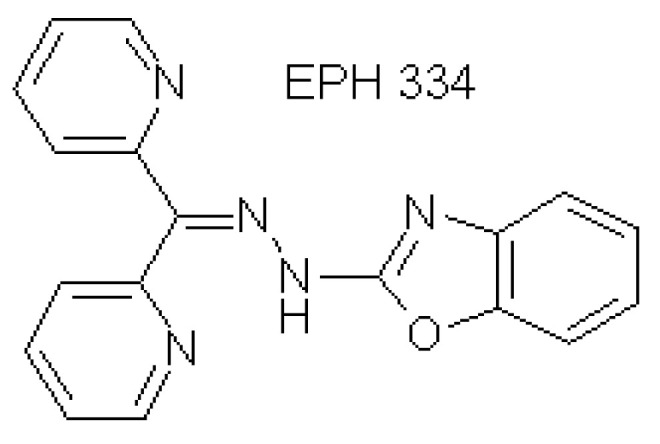
EPH334 chemical structure.

## Data Availability

Primary uncropped images and gel blots can be found online at MDPI (“Please notes all images were sent directly to the Editorial Office as requested, so they may ahev to deposit and provide a link”).

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
