# Peer review of "Host Cell Redox Alterations Promote Latent HIV-1 Reactivation through Atypical Transcription Factor Cooperativity"

_viruses, 2022, doi:10.3390/v14102288_

Round 1

Reviewer 1 Report

In this manuscript, the authors screened the small compound library for the potential usage as HIV-1 latency reversal agents (LRAs). By comparing with the reported control cytokine, TNFα, in T cell line latency model J-Lat 10.6, some compounds have been selected especially the most promising EPH334, which was subject to extensive investigations for its potency to reactivate HIV-1 from latency, including the evaluation for HIV-1 RNA processing (transcription) and ex vivo study with aviremic patient samples. Mechanism studies have revealed that EPH334 may reprogram the cellular ROS which triggers the activation of HIV-1 through MEK-ERK1/2 and HIF-1α pathways. Overall, this study was well-designed, figures were well-prepared and manuscript was well-written. Practically, the newly discovered LRAs may be considered for in vivo or (pre-) clinical trials to optimize “shock and kill” strategy.

I’ve listed some comments below, mostly are minor points which need to be clarified before the consideration of publication.

1 Based on your RNA-seq data and following mechanism studies, the EPH334-induced oxidative stress and downstream responses start very early (within 2h) after drug treatment, however, the HIV-1 reactivation was usually detected at 24h after treatment by qPCR and flow cytometry. Fig 2h showed that early timepoints (within 4h) have extremely low HIV-1 reactivation. Please explain the gap between oxidative stress and HIV-1 reactivation.

2 No Excel files about Supplementary Table 1-3 can be found, as mentioned in manuscript. Please confirm the upload or deposit of these files during resubmission.

3 There is a typo in line 601. Fig 6c should be Fig 6e.

4 The western blot of Fig 6a-b and Fig 6 c-d lack of internal control, such as actin. Same for Supplementary Fig 8.

Reviewer 2 Report

Cruz-Lorenzo et al describe a new chemical screen for latency reversing agents, focusing on derivatives of JIB-04. They identify a compound -EPH334 that has latency reversing activity in cell line models, a primary cell model, and in patient-derived cells.  They show through a series of experiments that this compound works by inducing ROS and activation of a kinase cascade leading to c-Jun/c-FOS activation and HIF1a stabilization.  Although the single agent activity of EPH334 is modest, this compound seems to combine with other agents to promote more efficient reactivation of HIV.  As such, this compound, and others like it could be a useful part of a combined LRA strategy for clinical clearance of the reservoir.  Overall the work is nicely done with appropriate models and controls, and will be of interest to researchers in the HIV latency field.  I only have a few minor points to make.

1. In Figure 2 the investigators look at the abundance of different HIV transcripts (US, SS, MS) after EPH334 dosing and argue that the splicing pattern is “correct”.  But I think it’s hard to make this claim without reference to a control – in this case, actively infected cells might be an appropriate comparison.

2. The figure legend to 2i indicates EPH334 concentrations of 0.5mM, 1mM and 10mM.  I suspect this is a typo, and that the correct units should be in uM?

3.  The reactivation data from patient samples is somewhat curious – one patient seems to respond potently to EPH alone but not in combination with JQ1.  Some discussion of this seems warranted.

4. Primary CD4 T cells would have been a better choice for the RNAseq experiment.  These data reveal that T cell activation pathways are increased, suggesting some effect on overall T cell activation.  It might be worth looking at surface activation markers (CD69, CD25, CD38, HLADR) after EPH334 stimulation to see what is happening.

5. Figure 4 - Since CellROX green emission spectrum overlaps with GFP, the control of unstained cells is a key one for this experiment.  This data is mentioned in the text but I couldn’t find this data in the manuscript. These data should be in a supplemental figure.

6. There are handful of typos and formatting issues that should be fixed.  Eg Line 476 – end of sentence seems to be missing? Line 799 – a word is missing here – should be “with SAHA”? Lines 951-952 – font here is different.

Reviewer 3 Report

Based on a previously identified latent CMV activator JIB-04, the authors identified JIB-04 and its analogues as potent inducer of HIV replication from latently infected cells. Further analysis of these compounds, named SMORE, revealed that SMOREs trigger cellular redox stress and induce the redox responsive program, resulting in the recruitment of redox-dependent transcription factors to the HIV promoter.  Authors conclude that manipulating cellular redox status represents a promising approach to reverse HIV latency. The results presented in the manuscript are very exciting and well-supported by the data. I recommend this manuscript to be published in Virus.

This is rather my curiosity and may be out of the scope of this study, but since SMOREs' HIV activation patterns seem to be very similar to LRAs that release P-TEFb from 7SKsnRNP (although they synergise with a P-TEFb releaser JQ1), I wonder if SMOREs also release P-TEFb from 7SKsnRNP. In addition, a recent paper from Karn indicate that MEK-ERK1/2 pathway plays an important role in P-TEFb biosynthesis (Mbonye PLOS Path 2021), which is a critical step for HIV activation in primary resting CD4+ T cells. If so, SMOREs have positive effects on multiple important pathways for HIV activation. Adding comments about it would make the manuscript even more exciting.
